# A massively parallel reporter assay library to screen short synthetic promoters in mammalian cells

Adam M. Zahm [1], William S. Owens[2], Samuel R. Himes[1], Braden S. Fallon [1], Kathleen E. Rondem[1], Alexa N. Gormick[1], Joshua S. Bloom[2,3], Sriram Kosuri[2], Henry Chan[2] & Justin G. English [1] ✉

Cellular responses to stimuli underpin discoveries in drug development, synthetic biology, and general life sciences. We introduce a library comprising 6144 synthetic promoters, each shorter than 250 bp, designed as transcriptional readouts of cellular stimulus responses in massively parallel reporter assay format. This library facilitates precise detection and amplification of transcriptional activity from our promoters, enabling the systematic development of tunable reporters with dynamic ranges of 50–100 fold. Our library proved functional in numerous cell lines and responsive to a variety of stimuli, including metabolites, mitogens, toxins, and pharmaceutical agents, generating robust and scalable reporters effective in screening assays, biomarkers, and synthetic circuits attuned to endogenous cellular activities. Particularly valuable in therapeutic development, our library excels in capturing candidate reporters to signals mediated by drug targets, a feature we illustrate across nine diverse G-protein coupled receptors (GPCRs), critical targets in drug development. We detail how this tool isolates and defines discrete signaling pathways associated with specific GPCRs, elucidating their transcriptional signatures. With its ease of implementation, broad utility, publicly available data, and comprehensive documentation, our library will be beneficial in synthetic biology, cellular engineering, ligand exploration, and drug development.

The cells' ability to receive stimuli is defined not only by the proteins involved but also cellular context, location, duration, integration, and parallel inputs from other signaling pathways. These signaling events culminate into cellular responses with significant molecular efficacy and frequently result in activation or regulation of the roughly 1600 transcription factors (TFs) encoded by the human genome. Transcription factors bind to specific DNA sequences known as transcription response elements (TREs) to initiate gene transcription[1–4]. Identical TREs can be found at multiple loci across the genome, serving as platforms to facilitate coordinated transcription programs for gene network activation[5]. The binding of TFs to TREs and their interaction with enhancers and other distal regulatory elements define how individual genes are regulated transcriptionally[6]. The aggregate change in cellular gene transcription is among the most commonly used indicators of cellular identity, state, and condition[7–9].

The TF-TRE mechanism has long been harnessed to create engineered systems. Genome-derived or synthetic promoters harboring TREs report on TF activity by driving transcription of user-selected genetic material. Promoters responding to specific cellular signaling events by modulating transgene expression in a reproducible manner

[1]Department of Biochemistry, University of Utah School of Medicine, Salt Lake City, UT, USA. [2]Octant Inc., Emeryville, CA, USA. [3]Department of Computational Medicine, University of California, Los Angeles, CA, USA. ✉e-mail: justin.english@biochem.utah.edu

are crucial for well-established applications (e.g., reporter assays) and emerging technologies, such as pluripotent stem cell lineage-control networks, regulatory control of human CAR-T cells in vivo, deep mutational scanning, and directed evolution[10–13]. Existing synthetic promoters often consist of TREs arrayed immediately upstream of an inactive or weakly functional minimal promoter. Transcription factor binding to the response elements activates the minimal promoter, initiating transcription. Due to their short length relative to endogenous promoter regions, synthetic promoters are advantageous for many applications, including those utilizing vectors with limited cargo sizes. However, off-the-shelf availability is extremely limited, often necessitating the in-house development of new promoters with desired characteristics (e.g., high dynamic range, defined basal activity, short length, etc.). The ability to quickly screen genome-derived or synthetic promoters suitable for specific applications at large scale will accelerate molecular tool development and deployment.

Technologies such as self-transcribing active regulatory region sequencing (STARR-seq) and massively parallel reporter assays (MPRAs) have been used to interrogate the effects of primary DNA sequence on gene expression at extremely high throughput[14–18]. These studies have finely mapped many rules governing the highly complex TF-DNA interactions at the heart of gene regulation. Here, we leverage the experimental principles of these high throughput formats to create a powerful tool to easily quantify responses of synthetic promoters that serve as downstream transcriptional readouts of specific upstream signaling events in mammalian cells. Our MPRA plasmid library can be used to survey over five hundred thousand barcoded plasmids representing 6144 synthetic promoters of less than 250 bp in length, containing candidate TREs derived from the ex vivo binding motifs of 229 human and mouse TFs. We demonstrate that single replicate transient transfections of this library can provide reliable transcription rate estimates for our episomal synthetic promoters across a range of stimuli, from heavy metal toxicity to G protein coupled receptor (GPCR) activation, and between numerous cell lines. Using this platform we determine the transcription factor signaling contributions of numerous GPCRs, notable drug development targets, discover distinct patterns of activity from each receptor, and derive

synthetic high dynamic range reporter constructs as readouts of their signaling activity.

## Results

### Identification of functional synthetic promoters using a massively parallel reporter assay

Coupling promoter activity to the production of barcoded mRNAs affords the ability to quantify activity in response to stimuli in a sensitive and high-throughput manner via next-generation sequencing (NGS)[15]. We developed a barcoded plasmid library (TRE-MPRA) of synthetic promoters composed of TF binding motifs derived from all unique DNA position weight matrices for hundreds of human and mouse TFs identified via HT-SELEX[19] (see Methods). We reasoned that TREs based on DNA sequences bound by TFs ex vivo would produce superior synthetic promoters compared to sequences based on genomic footprints and removed from their native chromatin context. Four copies of a given binding motif were arranged in six configurations (TRE units) and positioned immediately 5′ to one of three minimal promoters to create short, synthetic promoters (hereafter promoters) driving expression of a protein coding transcript (Luc2) with a barcoded 3′ UTR (Fig. 1, Supplementary Fig. 1). We also included hundreds of negative control promoters containing TREs of scrambled sequences in the library. Four independent plasmid library preparations from distinct liquid bacterial cultures were sequenced on separate flow cells and showed nearly identical barcode representation per promoter, as well as highly correlated barcode reads per million (Supplementary Fig. 2A, B). The mean and median barcode representation for promoters in a representative plasmid library preparation were 82 and 65, respectively. Of the 6318 promoters we designed, a total of 6144 (97%) were detected in our plasmid preparations.

We first benchmarked this library by transfecting the human HEK293 cell line and culturing the cells in serum-free media for 24 h to establish baseline transcriptional activities of each promoter. We quantified barcoded mRNA levels via NGS and then calculated the aggregate ratio of RNA to DNA reads per million as a proxy estimate of promoter transcription rate. Each of two independent experiments, both comprising four independent replicates of transfected cells and

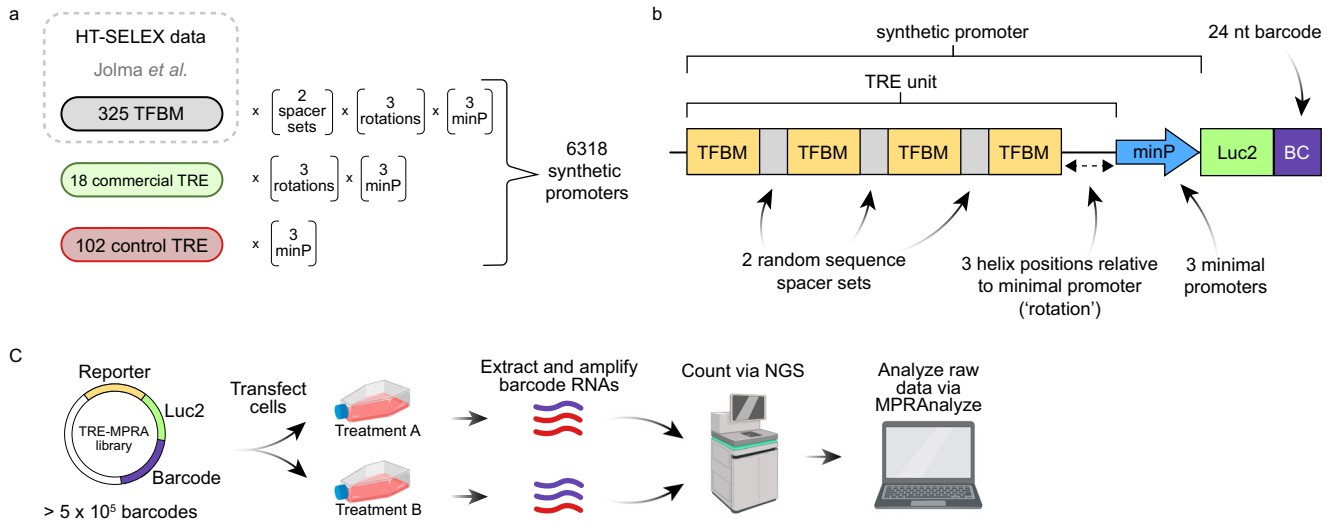

**Fig. 1 | TRE-MPRA overview. a** Transcription factor binding motifs (TFBMs) included in the TRE-MPRA library were derived from the HT-SELEX dataset of Jolma et al. Candidate synthetic promoters were designed by combining homotypic TFBMs and one of three minimal promoters in multiple configurations. Commercially available TREs, as well as negative control promoters were also included in the TRE-MPRA library. **b** Four copies of a given TFBM were oriented on alternating sides of the DNA double helix using spacer sets of random nucleotides and rotated along the helix relative to the minimal promoter (TRE unit). TRE units combined with

minimal promoters ('promoters') regulate the expression of a Luc2 CDS containing 24 nucleotide barcodes in the 3′ UTR. Barcodes were mapped to associated TRE units using next-generation sequencing (NGS) during library preparation.
**c** Barcoded RNAs were extracted from transfected cells and sequenced via NGS alongside the input TRE-MPRA plasmid libraries. Resulting barcode counts were used to generate estimated transcription rates and analyzed via MPRAnalyze to compare promoter activities between treatments. Created in BioRender. Zahm, A. (2024) https://BioRender.com/i28a348.

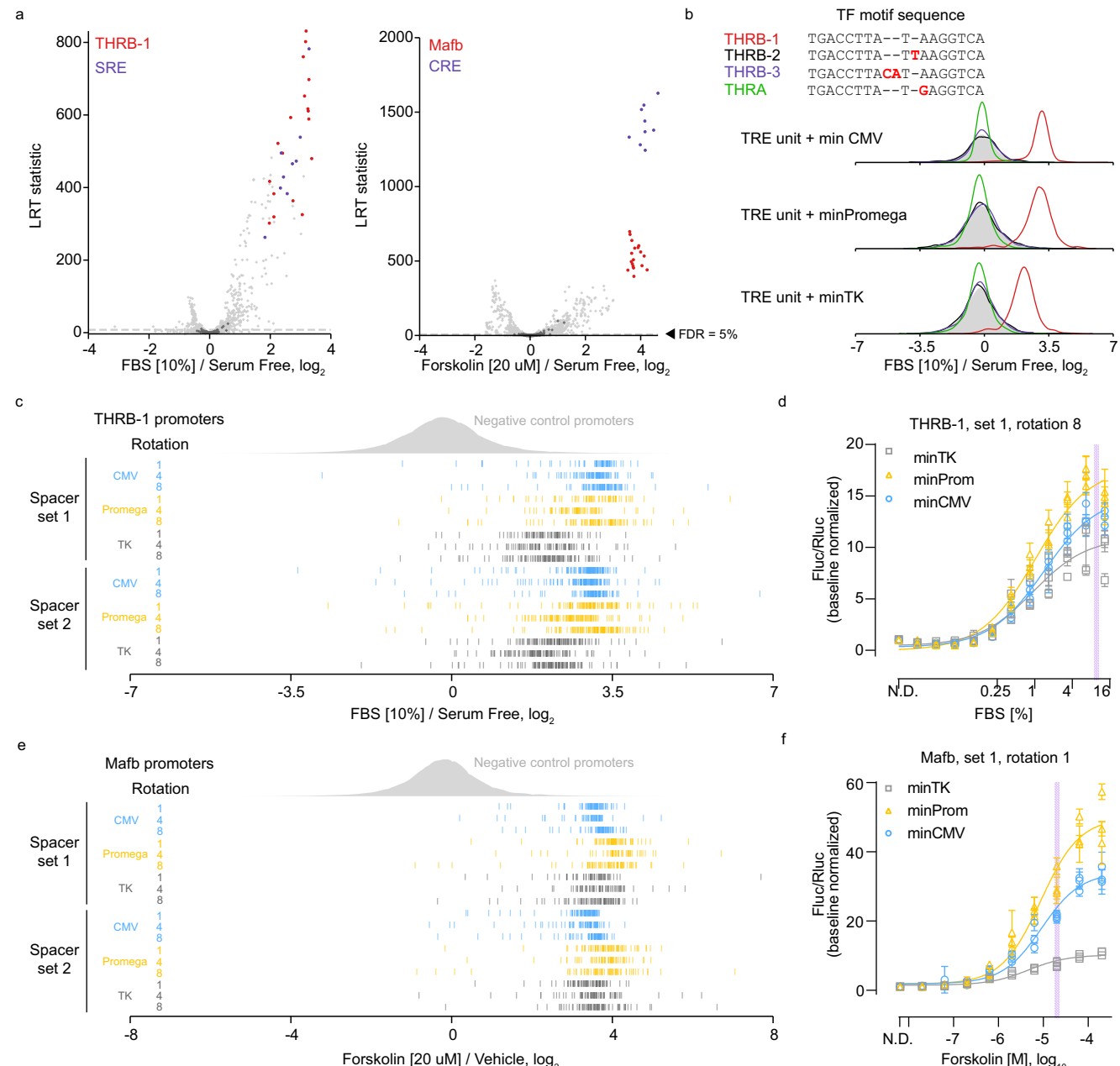

**Fig. 2 | TRE-MPRA benchmarking with fetal bovine serum and forskolin.**
**a** Promoter responses of HEK293 cells treated with 10% FBS (left) or 20 µM forskolin (right) ($n = 3$ each) compared to untreated cells ($n = 4$). Select promoters are colored by TFBM. Dashed lines: 5% FDR threshold. Dark gray data points: negative control promoters. LRT, likelihood ratio test. **b** Barcode fold changes for promoters containing the THRB-1 motif or similar motifs following 10% FBS treatment. Individual barcode responses of each THRB-1 (**c**) or Mafb (**e**) promoter in FBS or forskolin treated cells, respectively, relative to controls. **d**, **f** Dose-response curves of THRB-1 (**d**) and Mafb (**e**) TRE promoters tested in dual-luciferase assays. Data were scaled to the Fluc/Rluc ratio in untreated cells (N.D. – no drug). Data points and error bars: mean and standard deviation ($n = 4$ technical replicates) within each experimental replicate ($n = 3$ independent experiments). Shaded lines indicate TRE-MPRA doses.

sequenced on separate flow cells, showed a range of transcriptional rate estimates greater than 300-fold across all promoters (Supplementary Fig. 2C). Furthermore, rate estimates were highly correlated across the independent experiments (Spearman's $p = 0.960$, $p < 0.0001$) (Supplementary Fig. 2D). Across the population of TRE units, we observed a pronounced effect of the paired minimal promoter on baseline transcription rates, whereas the spacer sequences between TF binding motifs and the distance between the TRE unit and minimal promoter did not globally alter transcription (Supplementary Fig. 3).

Next, we transfected HEK293 cells with the TRE-MPRA library and subsequently treated the cells with fetal bovine serum (FBS) or

forskolin for six hours in triplicate. Cellular responses to serum and forskolin are classically associated with the serum response element (SRE) and cAMP response element (CRE), respectively[20-22]. Differential promoter activities relative to vehicle-treated cells transfected with the library were determined using MPRAnalyze[23]. As expected, serum treatment elevated expression from SRE promoters, while forskolin activated CRE promoters (Fig. 2a). Transcriptional activities from promoters containing one of three human thyroid hormone receptor beta (THRB) motifs (THRB-1) were significantly elevated in cells treated with FBS, whereas motifs of highly-similar sequence did not alter transcription in response to FBS (Fig. 2a, b). Notably, each of these sequence-similar but non-responsive promoters lacks a palindromic

structure like the responsive THRB-1 motif. We also observed increased activities from promoters containing the murine v-maf musculoaponeurotic fibrosarcoma oncogene family, protein B (Mafb) binding motif following stimulation with forskolin (Fig. 2a). Of note, the THRB-1 motif is highly similar in sequence to the CArG box of SRE bound by serum response factor (SRF) and thus may provide a readout of SRF activity rather than THRB[22]. Likewise, the Mafb motif (TGCTGACGTAAGCA) in the TRE-MPRA library contains a sequence very similar to the cAMP responsive element (TGACGTCA) and so may be activated by cAMP responsive element binding protein 1 (CREB1) as opposed to MAFB[21]. Importantly, the responses of these promoters were uncorrelated with barcode abundance in the plasmid library (Supplementary Fig. 4A). Both treatment conditions showed differential effects of spacer sets and the position of TRE units in the DNA helix relative to the minimal promoters for specific TF motifs (Supplementary Fig. 4B, C). This result suggests that, while not an important consideration for many TREs, the distance between TRE and paired minimal promoter significantly affects synthetic promoter utilization by certain TFs.

For each barcode, we calculated the median reads per million across biological replicates and then compared barcode fold changes (treatment versus unstimulated) of all THRB-1 and Mafb promoters against the panel of negative control promoters (Fig. 2c, e). Consistent responses across the populations of barcodes suggested the induction of individual promoters following stimulation was highly reproducible. We also noted the degree of induction of THRB-1 and Mafb TRE activity was dependent upon the paired minimal promoter (Supplementary Fig. 4D). To test these findings in an orthogonal assay, we derived dual-luciferase reporter plasmids containing individual promoters controlling the expression of a luc2P CDS, as well as a constitutive SV40-driven *Renilla* luciferase cassette. The promoters selected for validation reflected the general MPRA responses of all promoters containing these TREs under these treatment conditions. HEK293 cells transfected with reporter plasmids showed dose-dependent increases in relative luc2P activity following stimulation with FBS or forskolin (Fig. 2d, f). Furthermore, minimal promoter-dependent responses were observed, largely in agreement with our TRE-MPRA results (Supplementary Fig. 5A). These results demonstrate that our synthetic promoters can function as dynamic transcriptional readouts of cell signaling.

These orthogonal dual-luciferase experiments also found baseline transcription rates of the THRB-1 and Mafb promoters in untreated cells to be in agreement with the estimated transcription rates in untreated cells (Supplementary Fig. 5B). To further determine if constitutive transcription rate estimates derived from our TRE-MPRA experiment are reliable predictors of expression output in orthogonal assays, we derived and tested dual-luciferase reporter plasmids containing promoters spanning a range of estimated transcription rates from untreated HEK293 cells (Supplementary Fig. 5C). This series of reporters produced luciferase activities in line with the MPRA transcription rate estimates, with the exception of the BHLHB3 motif-containing promoter (Supplementary Fig. 5D). This discordant finding may reflect TF/TRE-specific translation rates observed in previous studies[24–26], and emphasizes the necessity to validate individual candidate promoters in orthogonal assays.

### Modulating additional synthetic promoters with additional stimuli

With a validated MPRA platform in hand, we next sought to modulate the activities of additional promoters in the library by treating HEK293 cells with eight additional stimuli, including mitogens and inducers of cellular stress (Supplementary Data 1). To increase throughput, we included a single replicate for most treatment conditions, as our preliminary results from FBS and forskolin treated cells showed that individual biological replicates identified the same sets of top

responding promoters as multiple replicates, albeit with lower precision (Supplementary Fig. 6). We performed hierarchical clustering to classify common and specific promoter responses (Fig. 3a). For most stimulus types, we observed hundreds of promoters with significantly altered activity, even at a stringent FDR cutoff of 5%, attesting to the statistical power of MPRAs. Across ten stimulus conditions, 1949 promoters (31.7%) showed altered transcriptional output in at least one condition relative to negative controls, while 207 promoters (3.4%) were altered in at least half of the conditions. Several treatments activated promoters with similar or greater effect sizes as was seen for FBS- or forskolin-responsive promoters. For example, dexamethasone treatment caused significant upregulation of TRE units containing motifs for two nuclear receptor superfamily class I (steroid) members: androgen receptor (AR) and glucocorticoid receptor (NR3C1) (Fig. 3b). A dual-luciferase plasmid containing one of our AR promoters showed a dose-dependent response to dexamethasone with a dynamic range of 134-fold (Fig. 3c). In addition, treatment with lithium chloride caused strong activation of TRE units with an NFAT5 or NFATC1 motif (Supplementary Fig. 7A), in line with lithium's inhibition of glycogen synthase kinase 3 beta, itself an inhibitor of NFAT transcriptional activity[27–31].

Our treatment conditions included two heavy metals: one physiologic (zinc) and one xenobiotic (cadmium). We noted that both metal treatments induced similar responses in metal response element (MRE)-containing and Tp53-containing promoters, relative to vehicle-treated controls, despite the apparent toxicity of zinc but not cadmium (Fig. 3d, Supplementary Fig. 7B). The MRE is bound by metal regulatory transcription factor 1 (MTF-1) in response to both zinc and cadmium elevation, as well as oxidative stress[32,33]. However, cadmium treatment showed two- to three-fold higher induction of HSF1- and heat shock element (HSE)-containing promoters relative to zinc. Previously, using a modified HEK293 cell line containing an HSE reporter, Steurer et al. observed[34] that HSE was roughly 200-fold more sensitive towards cadmium than zinc[35]. Because HSE is bound by both MTF-1 and HSF1, the observed HSE and HSF1 promoter activation discrepancies between zinc and cadmium treatments are likely a result of HSF1 activity rather than MTF-1[36].

Next, we transfected the TRE-MPRA library into additional mammalian cell lines in order to compare baseline transcription rates between cell lines from different species and tissue origins. While HEK293 experimental replicates had highly-correlated baseline transcription rates, correlations between different cell types were much lower (Supplementary Fig. 8A). To identify differentially active promoters between cell types, we generated biplot displays using standardized aggregate RNA to DNA ratios of barcode reads per million[37] (Fig. 3e). Several TRE units were highly active in subsets of cell lines, suggesting certain transcription factors have cell-specific basal activities (Supplementary Fig. 8B). Meanwhile, the basal transcription rates from Tp53-3 containing promoters in Neuro-2a and MDA-MB-231 cell lines were comparable to negative control promoters, whereas all other cell lines displayed much higher transcription rates (Supplementary Fig. 8B). Both Neuro-2a and MDA-MB-231 cell lines are known to harbor missense mutations in the DNA-binding region of p53 (V170L and R280K, respectively)[38,39], which may explain the lack of transcription from these promoters, as p53 missense mutations have been shown to hamper utilization of response elements in human p53 target genes[40].

To determine if these additional cell lines respond to serum treatment by differentially activating specific promoters, we transfected the cell lines with the TRE-MPRA library and treated them with FBS for 6 h. Serum responses were surprisingly discordant between cell lines, regardless of species (Supplementary Fig. 8C). For example, none of the additional cell lines significantly induced SRF units paired with minPromega or minTK minimal promoters, including those units based off of commercial reporters, whereas these promoters were

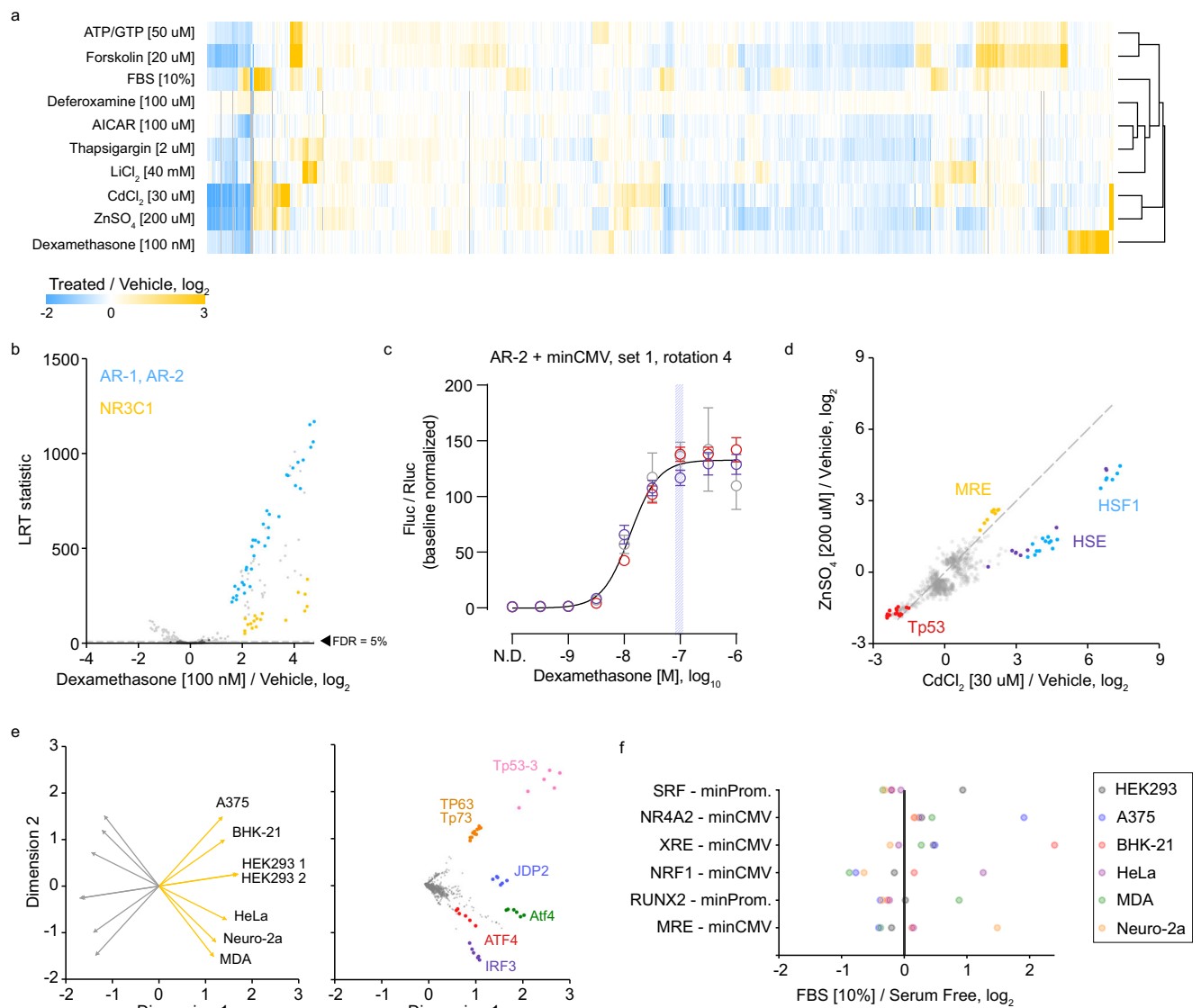

**Fig. 3 | TRE-MPRA results using additional stimuli. a** Heatmap of fold change responses in HEK293 cells across ten stimulus conditions relative to untreated cells. Treatment conditions were clustered using Euclidean distance with complete linkage. Displayed are the fold change values for all promoters that were significantly altered by treatment in at minimum one comparison. **b** Volcano plot of promoter responses following treatment of HEK293 cells with 100 nM dexamethasone in comparison to untreated cells. Dashed line indicates an FDR threshold of 5%. Negative control promoters are indicated by dark gray data points. **c** Dose response curve from HEK293 cells transfected with an AR promoter dual luciferase reporter and treated with dexamethasone (AR, minCMV, rotation 4, spacer set 1). Data were scaled (baseline normalized) to the Fluc/Rluc ratio in untreated cells (N.D. – no drug). The curve was fit to baseline normalized Fluc/Rluc values. Data points and error bars indicate the mean and standard deviation of technical replicates within each of three experimental replicates ($n = 3$ independent experiments, distinguished via color). The shaded vertical line indicates the dose of dexamethasone used in the TRE-MPRA experiment. **d** Scatterplot comparing promoter responses between lithium and cadmium treatments. HSF1 units were more responsive to cadmium. Dashed line indicates the identity line (y = x). **e** Separated biplots of standardized baseline transcription rates in the six mammalian cell lines. Displayed are the untreated cell line (left) and promoter (right) projections across two dimensions. All colored units are paired with the minCMV promoter. Orange and gray projection lines indicate the positive and negative directions of the treatment projection, respectively. **f** Example cell line-specific promoters responsive to treatment with 10% FBS. Each displayed promoter was significantly induced by FBS treatment in exactly one of six mammalian cell lines (FDR < 5%).

consistently induced in HEK293 cells (Fig. 3f). Indeed, each cell line responded to serum by activating subsets of promoters not activated in other cell lines. Because baseline promoter transcription rates and the responses to serum were discordant between cell lines tested, we recommend performing the TRE-MPRA screen in specific cell models of interest to identify optimal promoters.

## Specific synthetic promoters activated by aminergic GPCR agonism

G-protein coupled receptors (GPCRs) are 7 transmembrane proteins that, upon binding extracellular ligands, catalyze the exchange of GDP for GTP in heterotrimeric G-protein complexes to induce cellular signaling activities. GPCRs are present in every cell in the body and are involved in all known biological process from the detection of light in the retina and neurotransmitter relays in the brain to immune cell antigen detection and bone growth. Therefore, the 800 GPCRs of the human genome have been fruitful targets for drug development, with roughly 1/3 of all FDA approved drugs targeting a GPCR. The totality of signal integration from GPCRs remains unresolved and varies among cells and signaling context and there remains an urgent needed to more completely characterize the molecular and cellular consequences of ligand-receptor interactions for GPCR drug development.

After identifying synthetic promoters with large dynamic range responses across multiple chemical and mitogen treatments, we examined whether the TRE-MPRA library possesses the sensitivity necessary to detect transcriptional responses following GPCR activation. A limited number of TREs have long been used as readouts for GPCR signaling events, particularly as part of bioluminescent sensors[41], but these remain insufficient to fully map the input-output relationship between receptor activation and global cellular response. GPCRs can activate over 300 independent G-protein heterotrimers as well as non-canonical effectors, a complexity that cannot be captured through single, or even several, biological readouts[42–44].

Here, we employed TRE-MPRA to profile transcriptional changes induced by activation of three well-characterized aminergic GPCRs. HEK293 cells were co-transfected with the TRE-MPRA plasmid library and a plasmid expressing one of three human GPCRs: β2-adrenergic receptor (β2AR), 5-hydroxytryptamine receptor 2A (5-HT$_{2A}$), or dopamine receptor D2 (D$_2$R). These receptors selectively couple to Gα$_s$-, Gα$_q$-, and Gα$_i$-containing heterotrimeric G protein complexes, respectively. Upon GPCR activation by receptor agonists, these G protein complexes initiate distinct downstream signaling cascades[45,46]. After six hours of receptor agonist treatment (β2AR: 1 μM epinephrine; 5-HT$_{2A}$: 100 nM 5-HT; D$_2$R; 1 μM dopamine), we harvested RNA and analyzed differential barcode abundance between receptor alone and receptor with agonist conditions (Fig. 4a). Serotonin treatment in cells expressing 5-HT$_{2A}$ induced transcription from many promoters, including those containing THRB-1 and SRE units. Epinephrine treatment in cells expressing β2AR triggered increased expression from promoters containing Mafb and CRE motifs, similar to what we observed in forskolin treated cells, as both forskolin and Gα$_s$ signaling stimulate the formation of cyclic AMP. Conversely, dopamine binding to D$_2$R activates Gα$_i$ signaling, which leads to the inhibition of cyclic AMP production[47–49]. Surprisingly, no promoters were differentially active upon D$_2$R agonism. To determine if this result reflects technical issues with dopamine treatment, we performed the TRE-MPRA assay on two additional GPCRs: dopamine D1 receptor (D$_1$R; 1 uM dopamine), which is also agonized by dopamine but couples to Gα$_s$, and the mu-opioid receptor (μOR; 100 nM morphine), which couples to Gα$_i$. Unlike D$_2$R, agonism of D$_1$R with dopamine resulted in differential promoter activities resembling those observed for Gα$_s$-coupled β2AR (Fig. 4b, Supplementary Fig. 9A). The activation of μOR with the agonist morphine resulted in just two promoters with FDR values below 5%, in line with our results for D$_1$R, suggesting that activation of Gα$_i$-coupled GPCRs does not regulate cellular transcription in HEK293 cells (Supplementary Fig. 9A).

Having detected transcriptional changes mediated by exogenous, overexpressed GPCR agonism, we next asked whether the TRE-MPRA platform can detect transcriptional changes due to endogenous GPCR agonism, as well as changes resulting from receptor overexpression in the absence of an exogenous ligand. To address this question, we focused on β2AR in HEK293 cells, which are known to express functional β2AR endogenously[50]. We co-transfected cells with the TRE-MPRA library and a control plasmid expressing GFP and then cultured the cells in serum-free media in either the presence or absence of 1 uM epinephrine for six hours. Epinephrine treatment alone resulted in elevated transcription from promoters that had responded to epinephrine treatment in the presence of overexpressed ADRB2, suggesting that this platform is sensitive enough to detect endogenous GPCR signaling following agonism (Supplementary Fig. 9B). Furthermore, the overexpression of ADRB2 in the absence of epinephrine treatment also caused increased transcription from these promoters, suggesting that our platform can detect basal constitutive signaling from plasmid-expressed GPCRs (Supplementary Fig. 9B). Promoters containing CRE or Mafb units were sensitive to endogenous and exogenous ADRB2 signaling to similar degrees across the three minimal promoter combinations (Fig. 4c).

## Synthetic promoter activation following agonism of non-aminergic and promiscuous GPCRs

Having profiled synthetic promoter activity following agonist treatment of canonical Gα$_s$- (β2AR, D$_1$R), Gα$_q$- (5-HT$_{2A}$), and Gα$_i$-coupled (D$_2$R) aminergic GPCRs, as well as μOR, and identified G protein-specific changes following activation, we next utilized the TRE-MPRA platform to profile activation of additional GPCRs. We first assayed two non-aminergic GPCRs: the adhesion class protease-activated receptor-1 (PAR1) and the recently de-orphaned succinate receptor (GPR91/SUCR1)[51,52]. Agonist treatment of cells overexpressing either of these GPCRs significantly upregulated many promoters that were also increased by 5-HT$_{2A}$ agonism, such as THRB-1 and SRE-containing constructs, suggesting that PAR1 and GPR91 induce transcriptional changes predominantly via Gα$_q$ signaling (Supplementary Fig. 9C). Indeed, PAR1 and GPR91 profiles showed higher correlation with 5-HT$_{2A}$ than with the other assayed aminergic GPCRs (Supplementary Fig. 9D), though not identical, in line with observations that PAR1 and GPR91 couple to Gα$_q$ and additional Gα[53–56].

Next, we profiled two additional non-aminergic GPCRs that can strongly couple to and activate multiple distinct G proteins in HEK293 cells: MAS related GPR family member X2 (MRGPRX2) and neurotensin receptor 1 (NTSR1)[44,57–59]. Similar to PAR1 and GPR91, MRGPRX2 activated a set of promoters also activated by 5-HT$_{2A}$, suggesting MRGPRX2 also preferentially activates Gα$_q$ in HEK293 cells (Supplementary Fig. 9E). In contrast, NTSR1 agonism led to the activation of promoters also activated by either the Gα$_s$-coupled aminergic GPCRs (β2AR and D$_1$R) or 5-HT$_{2A}$ (Supplementary Fig. 9E). To compare the profiles of all nine GPCRs assayed, we generated biplot displays using the set of promoters that showed a significant response to receptor agonism in at least one of the nine datasets (Fig. 5a). Projections for PAR1, GPR91, and MRGPRX2 conditions were closely related to that of 5-HT$_{2A}$, again suggesting these GPCRs activate similar signaling pathways. Furthermore, projections of the Gα$_s$-coupled β2AR and D$_1$R were highly similar, while the Gα$_i$-coupled μOR and D$_2$R receptors showed minimal projections. In general, NTSR1 agonism resulted in the activation of both Gα$_s$- and Gα$_q$-responsive promoters (Fig. 5b), as denoted by a biplot projection located between β2AR and 5-HT$_{2A}$ and by correlation analysis (Supplementary Fig. 9D).

We next determined whether agonism of NTSR1 in the presence of the Gα$_q$-specific inhibitor FR900359 (hereafter inhibitor) would block the activation of promoters associated with 5-HT$_{2A}$ biplot projection. As expected, Gα$_q$-specific promoters such as NFKB1 were no longer activated by neurotensin treatment in cells pretreated with inhibitor (Fig. 5c, d). We noted that 5-HT$_{2A}$ agonism (Gα$_q$ signaling) significantly induced promoters containing CRE units, albeit to a lower magnitude than by Gα$_s$-coupled β2AR and D$_1$R receptor activation. NTRS1 agonism in the presence of inhibitor partially blocked the activation of CRE promoters, again suggesting that CRE activity, a canonical readout of Gα$_s$-coupled GPCR activation, is also induced by Gα$_q$ signaling (Fig. 5c, d). Surprisingly, inhibitor treatment did not significantly alter the activation of THRB-1 promoters, despite these being associated with the 5-HT$_{2A}$ projection (Fig. 5a–d, Supplementary Fig. 9F). We hypothesized that NFKB1 and THRB-1 promoters are readouts of distinct signaling pathways downstream of GPCR activity. To test this notion, we performed dual-luciferase reporter assays in HEK293 cells overexpressing 5-HT$_{2A}$ or NTSR1. Agonism of NTSR1 or 5-HT$_{2A}$ activated the NFKB1 promoter in a dose-dependent manner, and these responses were abolished in the presence of inhibitor (Fig. 5e). Similarly, the THRB-1 promoter responded to 5-HT$_{2A}$ agonism in a Gα$_q$-specific manner. However, as observed in the MPRA experiment, inhibition of Gα$_q$ did not block the response of the THRB-1 promoter to NTSR1 agonism, suggesting alternative signaling cascades activated by NTSR1 can induce THRB-1 promoters.

Finally, we noted that activation of AP1 units by GPCR agonism was strictly dependent on pairing with the minCMV promoter (Fig. 5b,

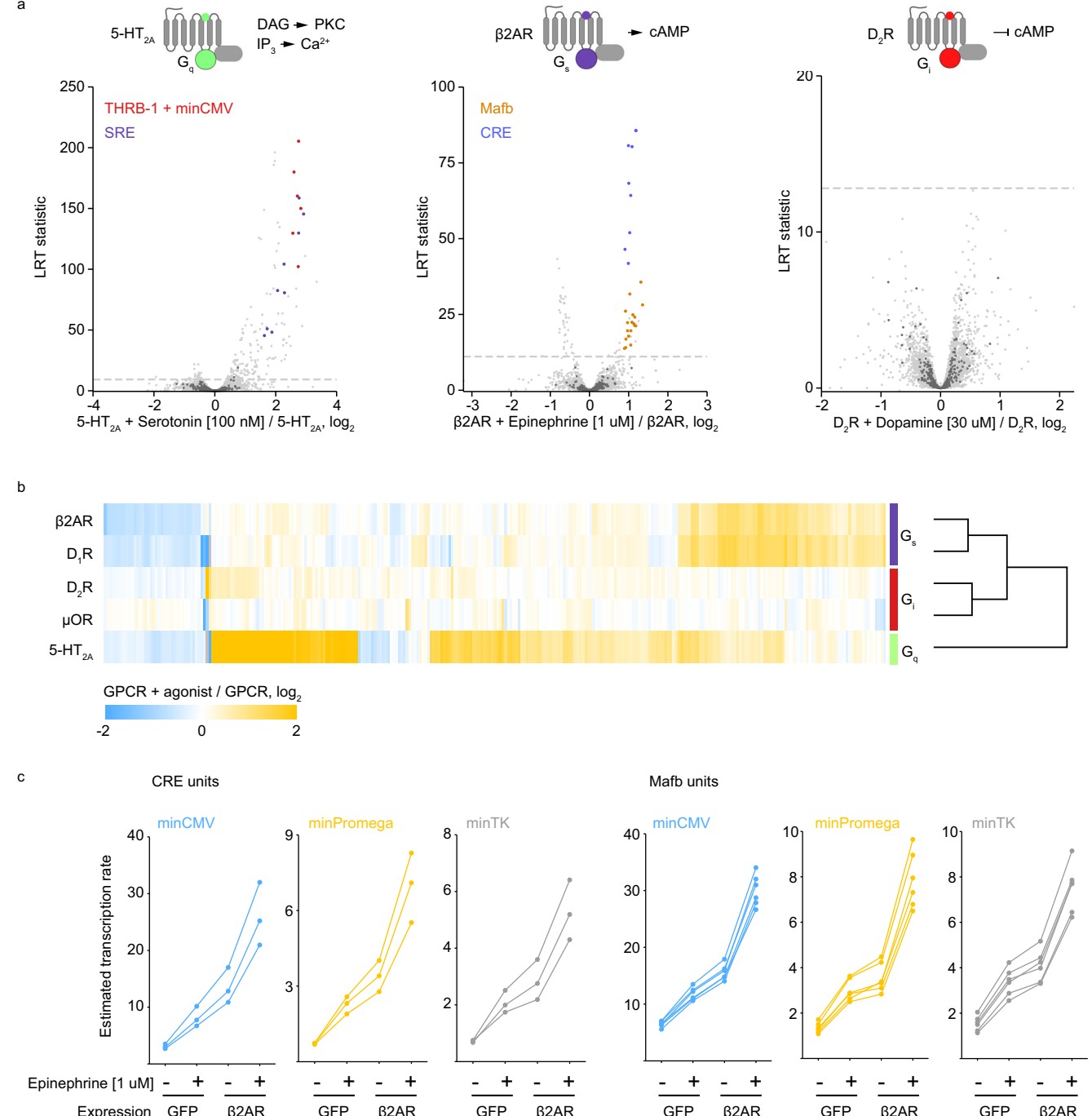

**Fig. 4 | TRE-MPRA detects distinct transcriptional signals downstream of aminergic GPCR agonism. a** Volcano plots of promoter responses following receptor agonism in HEK293 cells co-transfected with the TRE-MPRA library and GPCR expression plasmids. Dashed lines indicate an FDR threshold of 5%. Negative control promoters are indicated by dark gray data points. Created in BioRender. Zahm, A. (2024) https://BioRender.com/f26t609. **b** Heatmap of promoter fold change responses to GPCR agonism in HEK293 cells. Displayed are the fold change values for all promoters that were significantly altered by agonist treatment in at minimum one comparison. GPCRs were clustered using Euclidean distance with complete linkage. **c** Transcription rate estimates (aggregate RNA/DNA ratios) for CRE and Mafb units in HEK293 cells following 1 μM epinephrine treatment or ADRB2 overexpression, or both. Each connected set of data points represents a single promoter.

To validate this finding, we generated dual-luciferase reporters that replicated the sequences of minCMV and minPromega versions of AP1 TRE units of the MPRA library. We also converted a commercially available AP1 luciferase reporter into a dual-luciferase reporter by replacing its hygromycin expression cassette with an SV40/Rluc cassette for direct comparison with our synthetic promoters. We then co-transfected HEK293 cells with a GPR91 expression plasmid and the AP1 reporters and measured firefly and *Renilla* luciferase activities after six hours of cis-epoxysuccinate treatment. We observed a dose-dependent increase in transcriptional output for the AP1 unit coupled with minCMV, whereas the minPromega coupling and the modified commercial AP1 reporter showed little or no response to cis-epoxysuccinate, in agreement with the results of the TRE-MPRA experiment (Supplementary Fig. 10B).

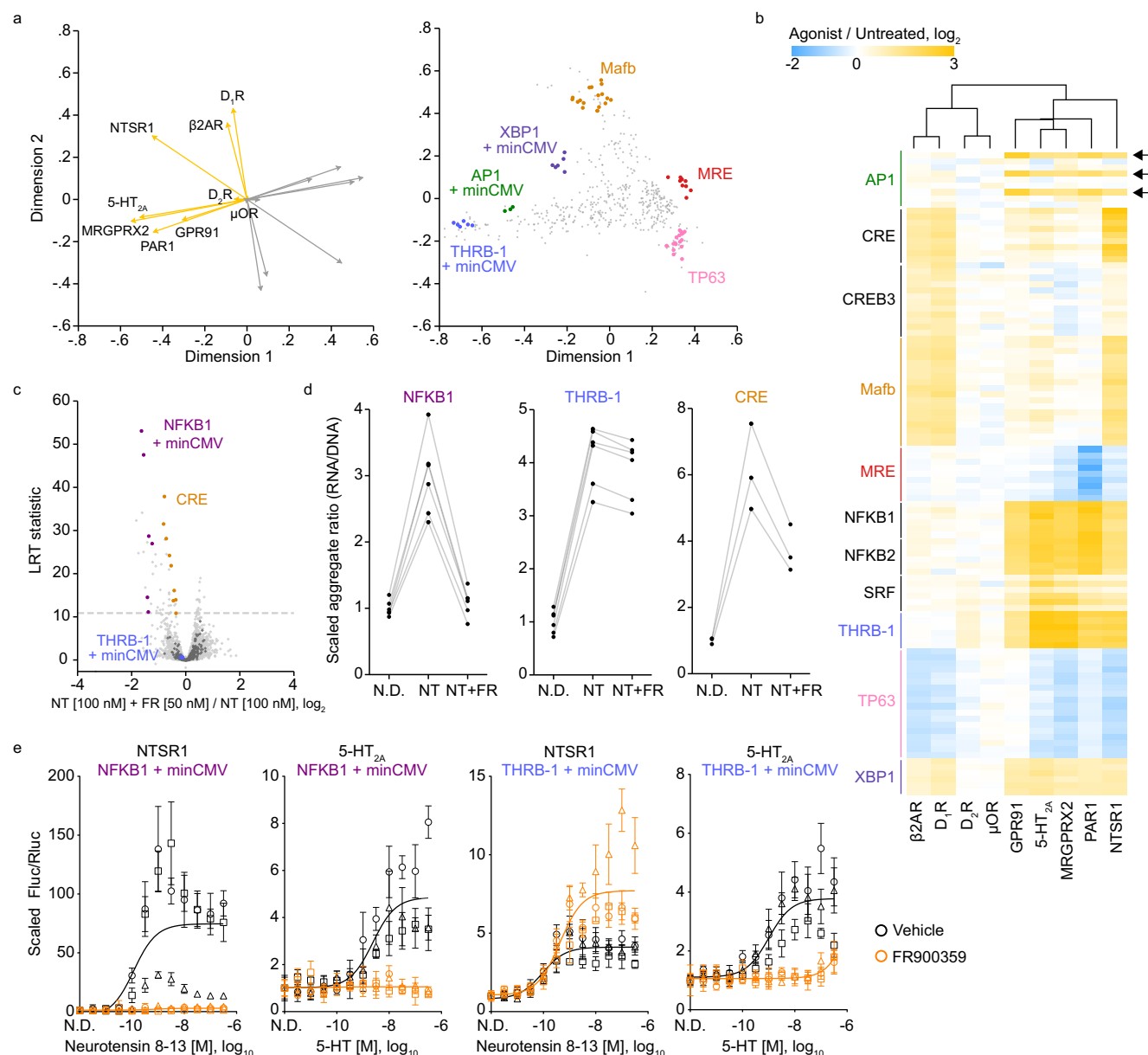

**Fig. 5 | Profiling promoter activity following activation of additional GPCRs.**
**a** Separated biplots for the set of promoters that showed a significant response to receptor agonism in at least one of the nine datasets. Displayed are the GPCR log$_2$ fold changes (agonist versus untreated) (left) and promoter (right) projections across two dimensions. Responding promoters of interest are highlighted. Orange and gray projection lines indicate the positive and negative directions of the treatment projection, respectively. **b** Heatmap of selected promoter responses to agonism across GPCR experiment groups. GPCRs were clustered using Euclidean distance with complete linkage. Arrows indicate the AP1 unit/minCMV promoters. A list of promoter (row) labels can be found in Supplementary Data 6. **c** Volcano plot of promoter responses following neurotensin 8–13 (NT) treatment in HEK293 cells co-transfected with the TRE-MPRA library and an NTSR1 expression plasmid, with or without addition of the Gα$_q$-specific inhibitor FR900359 (FR). **d** Scaled transcription rate estimates (aggregate ratios) for NFKB1, THRB-1, and CRE units paired with minCMV in HEK293 cells expressing NTSR1. N.D., no drug; NT, neurotensin 8–13 [100 nM]; NT + FR, neurotensin 8–13 [100 nM] + FR900359 [50 nM]. Each set of connect dots represents a single promoter. **e** Dual luciferase response curves in HEK293 cells in the presence or absence of 50 nM of the Gα$_q$-specific inhibitor FR900359 (THRB-1, minCMV, rotation 8, spacer set 1; NFKB1, minCMV, rotation 8, spacer set 1). Data were scaled (baseline normalized) to the Fluc/Rluc ratio in untreated cells (N.D. – no drug). The curve was fit to baseline normalized Fluc/Rluc values across three experimental replicates. Data points and error bars indicate the mean and standard deviation of four technical replicates within each of three experimental replicates (*n* = 3 independent experiments, distinguished via shape).

Distinct from the significant potential of this toolset for use in GPCR applications, we also demonstrated its utility in identifying compact synthetic promoters with high dynamic-range responses to a variety of mitogens and cellular stimuli. From these we created and validated a robust reporter for androgen receptor activity (Fig. 3c) and distinct heavy metal responsive reporters (Fig. 3d). In addition, we demonstrated that individual cell lines produce distinct basal and stimulus-dependent TRE-MPRA signatures, highlighting the utility of this biologically-responsive reporter library for use in cell-type specific applications. Our TRE-MPRA platform can be applied to aid in meeting the current significant demands for compact cell-type specific, stimulus-dependent, and combination reporters for synthetic biology applications in complex systems, as our suite of responsive elements represent some of the smallest, endogenously-coupled functional promoters with the highest transcriptional dynamic range yet generated for these applications[60–63].

## Discussion

Incorporating promoters with desired response profiles in reporter constructs enables precise monitoring of cell signaling and the development of synthetic biology applications where programmable transcription readouts are essential. This study introduces an MPRA library designed to simultaneously screen thousands of candidate synthetic promoters in mammalian cells. Using this library, researchers can easily identify promoters active in diverse cell lines and under various stimuli. This system can help streamline the development of complex gene regulatory networks and offers a valuable tool for the rational design of synthetic gene circuits with stringent input-output relationships.

We orthogonally tested several synthetic promoters in dual-luciferase assays and observed dose-dependent responses to stimulus, with dynamic ranges of 15- to 134-fold relative to baseline. Because our screening library contains only a handful of configurations for each candidate TRE, and all promoters were built with homotypic TREs, there is likely considerable room for optimization to further enhance performance upon orthogonal validation. Indeed, heterotypic TRE combinations with elevated transcriptional activity relative to homotypic promoters have been reported[64].

The MPRA format enables the simultaneous measurement of multiple technical replicates of the same genetic sequence of interest, which provides a high level of statistical power to detect even minor effect sizes. Our experiments consistently detected promoters with effect sizes of less than 15% between treatment groups while maintaining false discovery rates below 5%. Therefore, the TRE-MPRA library can effectively capture changes in promoter activity that might be missed by other screening formats. It is worth noting that the most suitable promoter for a given application might not be one of the largest responders in a specific screen. As a demonstration of the TRE-MPRA library's sensitivity, we were able to detect changes in promoter activity as a result of the agonism of endogenous β2AR in HEK293 cells. Furthermore, the overexpression of β2AR in the absence of ligand activated these same promoters. We anticipate the library becoming a powerful tool for studying GPCR signaling. GPCRs have traditionally been associated with activating Gα, β, γ transducers and arrestins; however, recent research has revealed signaling properties beyond these paradigms, including coupling to non-G-protein elements or inducing cellular signaling from endosomal compartments[65-69]. By profiling promoter activity changes in response to GPCR activation, we expect this system to help uncover novel regulatory elements and pathways of GPCR signaling.

Although we have demonstrated the value of our library in multiple contexts, several limitations inherent in our library composition warrant consideration. First, the library almost certainly is not able to capture the activity of every mammalian transcription factor. The HT-SELEX study upon which our synthetic promoters was based did not obtain position weight matrices for every human and mouse transcription factor and so our library will not provide direct readouts of their activities. It is plausible that synthetic promoters simply cannot be derived to detect certain TFs, which may lack functionality except upon native chromatin. In addition, our library may not include the spatial architectures necessary for certain TFs to regulate transcription from the minimal promoters. Promoter architecture diversity is a delicate design consideration, as including too few varieties may inadvertently render the library insensitive to certain TFs, whereas including more increases the library diversity and subsequently the experimental scale necessary to obtain library coverage without a similar gain in information. Our library has likely skewed toward the latter, as many TREs displayed a similar response across each configuration. Furthermore, as exemplified by our Gα$_i$-coupled GPCR activation experiments, certain biological responses will not be reflected by changes in the output of our synthetic promoters. One inherent shortfall in the MPRA format is lack of temporal information, as each sample will include barcodes expressed beginning from the time of transfection up to the time of collection. Because of the costs associated with NGS, we recommend temporal resolution be expanded during promoter validation.

Nevertheless, this library provides a resource to identify potential endogenously-coupled synthetic reporters for all experimental designs compatible with plasmid delivery and barcode recovery. We anticipate this system being of significant utility in applications where synthetic circuit design, drug target identity, or biological effectors of cellular response are required, but unknown, due to novel biological context or condition. Users can directly screen this library, identify reporters ready for immediate use, and leverage them for a myriad of applications including deep mutational scanning, preclinical therapeutic monitoring, and the control of chimeric antigen receptor T cells in vivo. The composition of our library enables users to select not only high fold-change reporters, but synthetic promoters from a wide range of sensitivities and amplitudes tuned for a given engineering application. This capability is particularly significant when viewed from the lens of a cellular synthetic biology. Our engineered reporters are activated by transcription factors and report on signal input at a layer in signal transduction independent of genomic context. This allows for fine-tuned selection of systems regulated by upstream signal inputs that impinge on transcriptional output in reliable and predictable ways. Furthermore, our MPRA plasmid library can be easily supplemented with new synthetic promoters based on the binding motifs of additional TFs, as warranted.

## Methods

### Reagents and drug preparation
FBS was purchased from Omega Scientific. Cis-Epoxysuccinic acid was purchased from ThermoFisher Scientific. Forskolin, dopamine, thapsigargin, and (R)-zn3573 were purchased from Tocris Bioscience. ATP and GTP were purchased from New England Biolabs. Neurotensin (8–13) (trifluoroacetate salt) and FR900359 were purchased from Cayman Chemical. (-)-Morphine sulfate pentahydrate was acquired from the National Institute on Drug Abuse Drug Supply Program. All other chemicals were purchased from Sigma-Aldrich. Dilutions of stock solutions were made in 3X drug assay buffer (0.3 mg/mL ascorbic acid, 0.3% bovine serum albumin, and 20 mM HEPES in HBSS).

### Transcription factor binding motif selection
TF binding motifs included in the initial screening library were selected from a published set of position weight matrices (PWMs) derived for 411 human and mouse TFs using HT-SELEX[19]. Beginning with the seed sequence of each PWM in the HT-SELEX dataset, we first trimmed fully degenerate nucleotides (Ns) from the 5' and 3' ends and then replaced all remaining degenerate nucleotides with the predominant nucleotides at those positions. Finally, to eliminate redundancy in our candidate list, we removed any resulting motif for which the entire sequence was represented within another motif, resulting in a total of 325 unique binding motifs (Supplementary Data 2). For example, if our position weight matrix processing produced the motifs CAAAAAC and AAAAA, we only designed promoters based on the CAAAAAC motif.

### TRE unit design
For each experimental binding motif, we designed six unique TRE units, each consisting of four copies of the binding motif separated by random nucleotide spacer sequences, a random sequence of variable length following the 3' motif, and flanked by restriction enzyme recognition sequences and primer binding sites (Supplementary Fig. 1). We chose to test homotypic TRE units containing four copies of the binding motifs based on a previous report showing maximal activity with synthetic promoters containing four copies of a cAMP response element[18]. Spacer lengths were selected based on individual motif lengths such that adjacent motifs were oriented on opposite

sides of the DNA double helix. To the resulting list of 1950 TRE units, we added 54 TRE units based on Promega's pGL4 Luciferase Reporter Vectors ranging in length from 160 to 194 nucleotides, and 102 negative control TRE units, for a total of 2106 TRE units (Supplementary Data 3). Each TRE unit was examined for compatibility with our restriction enzyme strategy and modified as necessary to remove unwanted recognition sites. All TRE unit oligonucleotides, except those encoding positive control TREs greater than 160 nucleotides, were synthesized as 160 nucleotides in length by adjusting sequence lengths 3' to the TRE unit to limit bias in PCR amplification prior to library assembly.

### Barcoding of TRE units

Synthesized oligonucleotides encoding TRE units were purchased from Twist Bioscience and pre-amplified via qPCR (Supplementary Fig. 1A, Supplementary Data 4). Following pre-amplification, barcodes were added to the 3' end of the amplicons using degenerate primers during 10 cycles of qPCR in six distinct reactions. Each reaction was processed separately for the remainder of the library generation procedure. Amplicons were digested with MluI and SpeI and ligated to pDonor_eGP2AP_RC (Addgene #133784) digested with MluI and SpeI in triplicate reactions. Each ligation reaction was transformed separately into NEB 10-beta competent cells in triplicate. Following transformation, triplicates were pooled and cultured in 2X YT media containing 50 μg/mL kanamycin at 30 °C with shaking at 225 rpm overnight.

To generate TRE unit/barcode dictionaries, dual-indexed amplicons were generated from the purified plasmid pools via PCR. We generated and sequenced two amplicons from each replicate using unique indices. Amplicons were sequenced on a NextSeq 550 (Illumina) using a NextSeq 500/550 High Output 300 cycle flow cell with custom sequencing primers (Supplementary Data 4). TRE unit and barcode sequences were extracted from demultiplexed read pairs. Resulting TRE unit sequences were compared to the expected sequences using Starcode[70]. TRE unit/barcode pairs for which the TRE unit was within a Levenshtein distance of two from the expected unit sequence were retained. Any barcode that was paired with multiple unique TRE units was discarded from the dictionary.

### TRE-MPRA plasmid library assembly

After generating the TRE unit/barcode dictionaries, fragments containing a firefly luciferase CDS and one of three minimal promoters – thymidine kinase (minTK), the minimal promoter of Promega's pGL4 plasmid suite (minProm), or cytomegalovirus (minCMV) – were created in a modified pDonor_eGP2AP_RC via Gibson Assembly, digested with KpnI and XbaI, and then ligated into the barcoded plasmid library replicates using the KpnI and XbaI restriction enzyme sites. Each replicate received one of the three minimal promoters.

Plasmid libraries for transfection were prepared by inoculating 100 mL of 2X YT media containing 50 μg/mL kanamycin with 200 μL of bacterial glycerol stocks and incubating at 30 °C with shaking at 225 rpm overnight. Cultures were pelleted and plasmids were purified using a Plasmid Maxiprep Kit (Qiagen) according to manufacturer's protocol. Plasmid preparations were then combined at equimolar ratios. TRE-MPRA libraries are available from Addgene under deposit number 82594.

### Cell culture and TRE-MPRA plasmid library transfection

All cell lines were acquired from American Type Culture Collection (ATCC; atcc.org): HEK-293 (CRL-1573), HeLa (CRM-CCL-2), MDA-MB-231 (CRM-HTB-26), A-375 (CRL-1619), Neuro-21 (CCL-131), BHK-21 (CCL-10). HEK-293, HeLa, MDA-MB-231, and A-375 cells were maintained in DMEM (4.5 g/L D-glucose) supplemented with 10% FBS, penicillin (100 IU/mL) and streptomycin (100 μg/mL) (hereafter growth media). Neuro-2a cells were maintained in EMEM supplemented with 10% FBS, penicillin (100 IU/mL) and streptomycin (100 μg/mL) (growth media).

BHK-21 cells were maintained in MEM α with GlutaMAX supplemented with 5% FBS, 10% tryptose phosphate broth, penicillin (100 IU/mL) and streptomycin (100 μg/mL) (growth media).

For experiments with the TRE-MPRA library, $5 \times 10^6$ cells were plated on 15 cm treated tissue culture dishes in growth media. The following day, cells were transfected with 10 ug of the TRE library ±5 μg of a GPCR expression plasmid using TransIT-2020 (Mirus Bio) according to manufacturer's instructions. After 6 h, cells were washed with serum-free (SF) versions of growth media (hereafter SF media) and then cultured overnight in SF media. The following day, stimulus was added to the media and cells were cultured for an additional 6 h. Cells were then trypsinized using 0.05% trypsin with EDTA, pelleted by centrifugation, and frozen at −80 °C until processing. Treatment conditions are listed in Supplementary Data 1.

### Nucleic acid isolation

RNA fractions from cell pellets were isolated using QIAshredder homogenizers and the AllPrep DNA/RNA Mini Kit (Qiagen). Immediately after homogenization, a pool of four synthesized spike-in RNAs (2.5 fM each, 10 fM total) was added to each 600 uL sample (Supplementary Data 4). Spike-in RNAs served to identify any samples with poor barcode recovery. RNA fractions were eluted in 30 μL nuclease-free water. Following isolation, the RNA fraction was treated with TURBO DNase (Invitrogen) to remove any plasmid DNA carryover. DNA removal from RNA fractions was confirmed by RT-PCR using the SuperScript IV One-Step RT-PCR System (Invitrogen) with and without addition of SuperScript IV RT Mix, followed by agarose gel electrophoresis. RT-PCR was performed according to manufacturer's instructions with the following conditions: 55 °C for 10 min, 98 °C for 2 min, 30 cycles of PCR (98 °C for 10 s, 60 °C for 10 s, 72 °C for 8 s), and 72 °C for 5 min. Primer sequences are listed in Supplementary Data 4.

### Sequencing library generation

Dual-indexed amplicons from RNA samples were generated using the SuperScript IV One-Step RT-PCR System (Invitrogen). 0.5−1.0 μL of RNA was used as the template in 20 μL reactions. RT-PCR was performed according to manufacturer's instructions with the following conditions: 55 °C for 10 min, 98 °C for 2 min, 17-27 cycles of PCR (98 °C for 10 s, 60 °C for 10 s, 72 °C for 8 s), and 72 °C for 5 min. Dual-indexed amplicons from plasmid DNA samples were generated using the Platinum SuperFI II Green PCR Master Mix (Invitrogen). PCR was performed according to manufacturer's instructions with the following conditions: 98 °C for 30 s, 16 cycles of PCR (98 °C for 5 s, 60 °C for 10 s, 72 °C for 5 s), and 72 °C for 5 min. Products were separated by agarose gel electrophoresis and library amplicons were extracted using the QIAquick Gel Extraction Kit (Qiagen). Amplicons were quantified with the KAPA Library Quantification Kit for Illumina Platforms (KAPA Biosystems) in 384-well format using a CFX Opus 384-Well Real-Time System (Bio-Rad). Libraries prepared from RNA samples were pooled at equimolar concentrations and combined with plasmid DNA input amplicons such that plasmid amplicons represented approximately 6−8% of the pool. RNA and plasmid DNA libraries were sequenced using a 50 cycle SP flow cell on a NovaSeq 6000 (Illumina) using custom sequencing primers (Supplementary Data 4).

### Processing of sequencing data

Raw barcode counts from TRE-MPRA samples were derived from demultiplexed fastq files by collecting the first 24 nucleotides of each read. Candidate barcodes were clustered within each sample via Starcode using a Levenshtein distance of one (starcode -i input_file.tsv --print-clusters -s -d 1 -o output_file.tsv)[70]. Reads of each barcode cluster were then cross-referenced with the barcode dictionary and only the candidate clusters with either (1) the centroid present in the dictionaries, and all other collapsed reads not present, or (2) the centroid present and any other collapsed reads present in the dictionary

mapping to the same synthetic promoter, were retained for analysis. Reads within each retained cluster were collapsed into a sum total for the cluster and assigned to the centroid barcode. Reads mapping to RNA spike-in sequences were also tallied within each sample. Spike-in reads were used to assess individual sample quality by flagging samples with disproportionately low proportions of TRE barcode reads; no such samples were observed in our datasets. Note: based on the barcode recovery and sequencing depth obtained in this study, we recommend using similar cell numbers, transfection methods, and sequencing library preparation methods, and to sequence at similar depth to help ensure data quality.

### Expression plasmid derivation

Plasmids used in this study are listed in Supplementary Data 5. pcDNA3.1_eGFP was generated by PCR amplifying the eGFP CDS from Arch(D95H)-eGFP (Addgene #51081) and inserting into pcDNA3.1(-)/*myc*-His A using the EcoRI and NotI restriction sites. pcDNA3.1_Signal-Flag-ADRB2 was generated by PCR amplifying the ADRB2 CDS from ADRB2-Tango (Addgene #66220) (introducing a stop codon) and inserting into pcDNA3.1(-)/*myc*-His A using Gibson assembly. pcDNA3.1_Signal-Flag-DRD2 was generated by PCR amplifying the DRD2 CDS from DRD2-Tango (Addgene #66269) (introducing a stop codon) and inserting into pcDNA3.1(+) using the NotI and XhoI restriction sites. pcDNA3.1_Signal-Flag-DRD1 was generated by PCR amplifying the DRD1 CDS from DRD1-Tango (Addgene #66268) (introducing a stop codon) and inserting into pcDNA3.1(-)/*myc*-His A using the BamHI and KpnI restriction sites. pcDNA3.1-GPR91 was generated by PCR amplifying the GPR91 CDS from SUCNR1-Tango (Addgene #66507) (introducing a stop codon) and inserting into pcDNA3.1(-)/*myc*-His A using Gibson assembly. pcDNA3.1_Signal-Flag-OPRM1 was generated by PCR amplifying the OPRM1 CDS from OPRM1-Tango (Addgene #66464) (introducing a stop codon) and inserting into pcDNA3.1(+) using the NotI restriction site. pcDNA3.1_Signal-Flag-NTSR1 was generated by PCR amplifying the NTSR1 CDS from NTSR1-Tango (Addgene #66457) (introducing a stop codon) and inserting into pcDNA3.1 using the HindIII and AflII restriction sites. pTwist_HA-PAR1 was purchased from Twist Bioscience. pcDNA3.1_Signal-Flag-MRGPRX2 and pcDNA5/FRT/Signal-Flag-HTR2A were previously described[13]. These two plasmids contain constitutive expression cassettes for the listed fusion proteins.

### Dual-luciferase TRE reporter plasmid derivation

The hygromycin CDS of pGL4.33[*luc2P*/SRE/Hygro] (Promega) was replaced with a *Renilla* luciferase CDS via Gibson assembly using NEBuilder HiFi DNA Assembly Master Mix (New England Biolabs) to generate a dual-luciferase reporter containing luc2P and *Renilla* expression cassettes (pGL4.33 R). To generate a TRE unit acceptor site upstream of the minimal Promega promoter replicating the promoter sequence of our TRE-minPro screening plasmids, annealed oligos containing restriction enzyme recognition sites were ligated to pGL4.33 R sequentially digested with BglII and KpnI (pGL4.R_TRE_minPro). To derive a minimal CMV promoter/TRE driven dual-luciferase plasmid, the minimal CMV promoter region of the TRE-MPRA minCMV plasmid was first PCR amplified, digested with ApaI and KpnI, and ligated to pGL4.33 R digested with ApaI and KpnI (pGL4.R_minCMV). Next, to replicate the promoter sequence of our TRE-minCMV screening plasmids, annealed oligonucleotides containing restriction enzyme recognition sites were ligated to pGL4.R_minCMV digested with KpnI (pGL4.R_TRE_minCMV). To derive a minimal TK promoter/TRE driven dual-luciferase plasmid, the TK promoter fragment of the TRE-minTK library was ligated to pGL4.R_minCMV digested with ApaI and KpnI (pGL4.R_minTK). Next, to replicate the promoter sequence of our TRE-minTK screening plasmids, annealed oligonucleotides containing restriction enzyme recognition sites were ligated to pGL4.R_minTK digested with KpnI (pGL4.R_TRE_minTK).

Annealed oligonucleotides corresponding to individual TRE units were ligated into pGL4.R_TRE plasmids digested with MluI and KpnI (minCMV and minPro) or AscI and KpnI (minTK) to generate dual-luciferase reporters with synthetic promoters identical to those of the MPRA constructs controlling expression of the luc2P CDS.

### Dual-luciferase assay

HEK293 cells were plated on 10 cm tissue culture-treated dishes in growth media. The following day, cells were transfected with 1 μg of TRE reporter plasmid using TransIT-2020 according to manufacturer's instructions. After 6 h, cells were detached from culture dishes with Dulbecco's phosphate-buffered saline containing 500 uM EDTA if co-transfected with a GPCR or with 0.05% trypsin otherwise, washed in SF media and plated on poly-d-lysine coated 384-well plates (10,000 cells/well in 20 μL of SF media). The following day, 10 μL of drug dilutions in SF media were added to wells and plates were incubated for six hours. Firefly and *Renilla* luciferase activities were then measured sequentially on a PHERAstar *FSX* (BMG Labtech) using the Dual-Glo Luciferase Assay (Promega, catalog #E2920) according to manufacturer's instructions.

### Transcription rate estimation and comparative analyses

Estimates of promoter transcription rates were calculated by summing the reads per million for all barcodes of a single promoter and dividing this result by the sum of the reads per million for all barcodes of the same promoter in the input plasmid library (aggregate ratio).

Comparisons of promoter activities between treatments were performed using MPRAnalyze version 1.22.0[23]. MPRAnalyze utilizes raw NGS read data of individual barcodes to perform comparisons of individual promoters between treatment groups. For each promoter with more than 100 associated barcodes, we selected the 100 barcodes with highest abundance in the plasmid DNA libraries for inclusion in these analyses to reduce computation overhead. All other promoters had all barcodes included in the analyses. Individual sample read depth factors were calculated by scaling the upper quartile of raw read counts to that of an arbitrarily chosen reference sample (upper quantile read counts/ reference sample upper quartile read counts).

To compare activity between treatment conditions, MPRAnalyze performs likelihood ratio tests (LRTs). Because many resulting $p$-values and false discovery rates were less than $10^{-38}$ and were thus outputted as zeros, we have chosen to display the resulting LRT statistic in our volcano plots. For those interested, the reported LRT statistics can be converted to $p$-values using a chi-square distribution with one degree of freedom, according to Wilks' theorem. The fitted models for comparative analyses were:

with biological replicates:

$$dnaDesign = \sim barcode + batch + condition,$$

$$rnaDesign = \sim condition$$

without biological replicates:

$$dnaDesign = \sim barcode + condition,$$

$$rnaDesign = \sim condition$$

MPRAanalyze fold change outputs were then converted from the natural log base to log base two.

### Statistics and reproducibility

One-way analysis of variance, Spearman's correlation tests, and Wilcoxon's rank-sum tests were performed using Sidak's adjustment for

multiple comparisons[86]. Heatmaps were generated using Morpheus[71]. Hierarchical clustering was performed via Morpheus using Euclidean distance with complete linkage. Dose response curves were generated using GraphPad Prism 9.5.0 using three or four parameter models. No statistical method was used to predetermine sample size. No data were excluded from the analyses. The experiments were not randomized. The investigators were not blinded to allocation during experiments and outcome assessment.

## Reporting summary

Further information on research design is available in the Nature Portfolio Reporting Summary linked to this article.

## Data availability

Source Data are provided with this paper. The raw sequencing data in this study have been deposited in the NCBI Gene Expression Omnibus under the series accession code GSE271608. Processed data presented in figures are provided in the Source Data file. Transcription rate estimations and pairwise sample comparisons can be explored online [https://jgenglishlab.github.io/mpra_vis.html]. Source data are provided with this paper.

## Code availability

Our TRE-MPRA analysis software is available [https://github.com/JGEnglishLab/TRE-MPRA-Pipeline]. https://doi.org/10.5281/zenodo.13905716[72].

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

## Acknowledgements

This work was supported by an award from the National Institute of General Medical Sciences (1DP2GM146247-01) to J.G.E.

## Author contributions

A.M.Z. designed the study, performed experiments, analyzed data, and wrote the manuscript. W.S.O. generated the TRE-MPRA plasmid library, performed experiments, and edited the manuscript. S.R.H. developed the analysis software, analyzed data, and edited the manuscript. B.S.F., K.E.R., and A.N.G. performed experiments and edited the manuscript. J.S.B. analyzed data and edited the manuscript. S.K. and H.C. designed the study and edited the manuscript. J.G.E. designed the study, analyzed data, and edited the manuscript.

## Competing interests

W.S.O., A.N.G., J.S.B., S.K., and H.C. hold equity in and/or are employed by Octant Inc., a company where similar types of assays are used for its drug discovery efforts. The remaining authors declare no competing interests.
