## [Transparent Peer Review file · Nature Communications]

A Massively Parallel Reporter Assay Library to Screen Short Synthetic Promoters in Mammalian Cells

Corresponding Author: Professor Justin English

Version 0:

Reviewer comments:

Reviewer #2

(Remarks to the Author)
Summary

In Zahm et al, 2024, the authors describe = development of a massively parallel reporter assay (MPRA) for screening a library of >6,000 transcriptional reporters consisting of a combination of transcription factor binding sites, spacer lengths, DNA helix rotations, and minimal promoters using RNA abundance measurements from barcoded reporters. The authors assay the library under different culture conditions, such as FBS and forskolin, and observe responses from TFs in the library that are consistent with existing literature. They also test the library in multiple cell lines, identifying TFs that generate cell-type/cell-line specific transcriptional responses. Finally, they demonstrate the therapeutic potential of the library by assaying it against several known GPCR + agonist combinations, finding variants that respond differentially to these targets, and elucidate discrete signaling pathways associated with specific GPCRs.

Overall, this manuscript is well-written, and the results are clearly presented, with rigorous technical analysis of their results and replicate experiments that demonstrate the robustness of their method. I think it is entirely appropriate for publication in Nature Communications. However, the authors should address the following comments before the manuscript is published:

Major Comments

1. The authors should clarify the language in the text describing the purpose and utility of their study. The abstract refers to the utility of a “massively parallel reporter assay platform”, that hints at a generalizability of their approach to other contexts. However, the text, and particularly the discussion, only argue for the utility the specific library used in the manuscript. When the authors describe their utility of their work, are they refereeing to the specific methodology used, i.e., cloning of a barcoded library of diverse transcriptional reporters and assaying using RNA sequencing, or to the specific set of variants (TFBSs, minimal promoters etc.) that they used in their experiments? If it is the former, the authors should describe how their platform differs from pre-existing methods (such as STARR-seq, exp-STARR-seq etc.) and what advantages it offers. If it is the latter, the language, particularly in the abstract, should be modified.

2. Related to point 1, the authors should provide a more comprehensive discussion of the library design choices they made. For instance, Lines 294-295 state, “Importantly, this platform enables a unique method for identifying endogenously coupled synthetic reporters for 295 any signal input”. While the authors describe developing their library from HT-SELEX, it appears they sampled only a subset of the variants from the cited paper. It is not clearly communicated how this subset was chosen. Moreover, it is unclear if these TREs are chosen for diversity of cell signal detection. If their approach is indeed generalizable to other library compositions and experimental contexts, it would be helpful to understand the motivations for how they chose their library and what classes of problems they expect their platform to address. It would be useful to see an overview, perhaps in figure form, that uses insights from their experiments to help the reader understand, e.g., how many TREs, min promoters, number of barcodes, sequencing depth, etc. should be considered for deployment of this platform in for other applications in other experimental contexts.

3. GPCR results observed in Fig. 4 should be contrasted/normalized with those observed in Fig. SB. In Fig 4, the library is transiently transfected along with a GPCR and cognate small molecule agonists, and the fold change in transcriptional

output is computed as the ratio of the GPCR + agonist counts over the GPCR counts alone. However, the results in Fig S9B show that there is an effect observed by simply treating the cells with the agonist without transfecting the cognate GPCR, which the authors attribute to endogenous GPCR signaling. Given this result and hypothesis, this effect should be subtracted from the values obtained in Fig. 4A to distinguish endogenous GPCR activation from the effects of the transiently transfected GPCRs to clearly distinguish the two.

Minor Comments

1. Fig. S2B does not mention which correlation coefficient is shown in the plots (I'm assuming it's Spearman's ρ), and should be added to the figure legend
2. Page 2, lines 82-83, "...whereas the spacer sequences between TF binding motifs and the distance between the TRE unit and minimal promoter did not globally alter transcription". The authors should mention if the parameters relevant to rotation/spacing they considered in this study were not sufficient to cause a significant change in expression, or that these features do not dictate transcription rate at large. The authors should also offer some discussion on the utility and importance of testing different rotations, spacers, etc
3. In Fig. 2D, F, why were the specific choices of rotations to plot used in these plots (ie rotation 8 for THRB-1, set 1, for 2D, and rotation 1 for MafB, set 1 for 2F)? Do the other rotations not show these effects, or was this one chosen at random?
4. Page 3, Line 117-118, "This discordant finding may reflect TF/TRE-specific translation rates observed in previous studies, and emphasizes the necessity to validate individual candidate promoters in orthogonal assays". This is a very salient and important point made by the authors. MPRA results should often be validated in a secondary experiment, especially when screening in a different context!
5. In Fig. 3D, S7B, when testing the library against heavy metals, were any cell health effects or toxicity observed? The authors should consider showing that there were no effects on cell health, especially due to there being only a single replicate used in these assays. If any effects of cell health are observed, the authors should comment on the same
6. Page 4, Line 179, potential typo - "single", instead of "signal"
7. Page 5, Line 267, "...promoters with the highest dynamic-range yet generated for these applications". This claim should be coupled with potential caveats of using a luciferase based reporter vs other, more commonly used outputs such as fluorescent proteins e.g. EGFP. Assays and resulting fold changes from an enzymatic reporter cannot be directly compared against FP based outputs due to the significant differences in their mechanism

Reviewer #3

(Remarks to the Author)

The author presents a highly efficient, massively parallel reporter assay platform of synthetic promoters, designed to assess cellular responses to stimuli quantitatively. As the author suggests, this platform enables precise transcriptional activity detection and amplification, facilitating the development of tuneable reporters with dynamic ranges of 50-100 fold. Demonstrating broad applicability, the platform functions across multiple cell lines and responds to various stimuli, including metabolites, mitogens, toxins, and pharmaceutical agents. Notably, the authors also demonstrate its utility in therapeutic development by generating unique reporters for drug target signals, particularly for G-protein coupled receptors (GPCRs), critical in drug discovery. The platform's ability to isolate and define specific signaling pathways associated with which could be valuable tool in synthetic biology, cellular engineering, ligand exploration, and drug development. Overall the paper is well written with well-designed and sufficient experiments but quite a few concerns need to be addressed.

The paper mentions existing technologies like STARR-seq and other MPRA's but does not adequately compare them to the new approach introduced in this study. A more detailed comparison would help in highlighting the novelty and potential advantages of the presented library.

The discussion does not address the potential limitations of the approach for the synthetic promoters being studied.

Acknowledging these upfront would provide a more balanced view of the research and its potential challenges.

The paper does not fully address potential non-specific effects of treatments, which could confound the results and compromise the reliability of the conclusions. Specifically, the study lacks controls for non-specific effects in experiments using fetal bovine serum (FBS), forskolin, zinc, and cadmium, potentially leading to misinterpretation of promoter activity changes.

For example: Line 84-100: The paper describes treating HEK293 cells with fetal bovine serum (FBS) and forskolin to benchmark the synthetic promoters but does not mention controls for non-specific effects. This lack of controls could lead to misinterpretation of the promoter activity changes.

Line 137-144: When discussing heavy metal treatments (zinc and cadmium), the authors note differential promoter responses but do not explicitly discuss controls for potential non-specific effects of these treatments.

Figures illustrating promoter responses to these treatments also omit necessary controls, such as vehicle-only conditions, which are crucial for distinguishing specific from non-specific effects. Also including antagonists for aminergic receptors as a parallel condition could have helped delineated non-specific further more. Including such controls and measuring off-target effects would improve the accuracy and interpretation of the synthetic promoter responses. In fact, for at least one of the GPCR a dose-dependent readout of the promoter activity readout also may better reflect usability of these tool sets.

The study's measurements were taken at a single time point, which may not fully capture the dynamics of GPCR signaling and promoter activation. This approach risks missing important transient or delayed responses, leading to an incomplete understanding of promoter activities. For instance, in line 75-80, The paper mentions transfecting HEK293 cells and measuring promoter activities after a specific treatment but does not detail multiple time points.

And in line 156-163, author mentions serum responses were measured at a single time point (6 hours) across different cell lines, revealing discordant responses but lacking temporal resolution. Comparing promoter rankings and responses to fetal bovine serum (FBS) also reflects this limitation. Including multiple time points in the experimental design would provide a more comprehensive view of promoter activation, revealing early, intermediate, and late transcriptional responses that will certainly improve the temporal resolution of GPCR signaling dynamics.

In Lines 198-199, the authors suggest that activation of G α i-coupled GPCRs does not influence cellular transcription in HEK293 cells. However, the lack of perturbation observed with the D2R receptor in terms of promoter activity may be attributed to the dopamine concentration used in the experiment, which was 1 μ M. It is plausible that a higher concentration could induce measurable activity, as the EC50 of the endogenous ligand for the D2R receptor is within a higher ligand concentration range. Thus, the potential for transcriptional regulation at increased dopamine levels cannot be definitively ruled out. Authors may try their toolset in other Gi coupled receptors and verify this claim.

In the experiments involving endogenous β 2AR, the observed increase in transcription from the promoters in the absence of epinephrine treatment was attributed to basal constitutive signaling, a well-documented phenomenon. While this is a plausible explanation, attributing the observed transcriptional activity specifically to autocrine signaling would be an overextension. To substantiate the claim of autocrine activity, the authors should provide supporting experimental evidence or reference similar findings from other studies. In the absence of such corroboration, it would be prudent for the authors to refrain from making this claim (line 207-208).

Version 1:

Reviewer comments:

Reviewer #2

(Remarks to the Author)

The authors have made significant changes in response to the comments I presented, improving the clarity of the manuscript, and clearly communicating the novelty of their method, which lies in the library they have developed as a means for identifying candidate transcriptional response element arrays for a range of downstream applications. They do a good job of clarifying that the library screen is meant to serve as a starting point for users, and that appropriate validation in the user-defined context of their application will be necessary to screen any "hits" returned from the initial screening of their library will be crucial, as it is an important point for most high-throughput assays used in this fashion.

Based on the authors' responses to the comments, their edits to the main text and supplement, I believe the manuscript is now fit for publication in nature communications. Detailed responses to their rebuttal of my specific comments are appended below:

Major Comments

1. The authors' edits to the abstract sufficiently address this comment and bring clarity to the value of their work, i.e. a large-scale exhaustive library of transcription factors, minimal promoters, and some other relevant features that is meant to serve as a first-pass screen for a variety of applications.
2. The additional context added by the authors to this claim sufficiently addresses this comment. The motif selection strategy that the authors used makes sense, and the changes they have made to the material, as well as to the methods section. However, the example they have presented here where instances of one motif being a subset of another and therefore being excluded clarifies their filtering approach very intuitively, and I'd urge the authors to include this example in their methods section as well, to provide further clarity to the readers. The significant edits made to the text describing potential use cases for their library also brings clarity. Further, the overarching clarification of the goal of this work being the development of this library, which wasn't immediately apparent during the initial submission coupled with these edits have simplified the nature of the discussion necessary with regard to this comment. If this work served as more general platform for RNA-sequencing based readouts of pooled variants using barcode libraries, there would be additional clarification necessary, but the authors have clarified that that is not the case with this work. The recommendations the author make in the methods section is an important addition to the manuscript. However, I'd encourage the authors to include specific numbers regarding their sequencing read depth and any other relevant metrics in their methods section, to gain further insight into the specifics of their results.
3. The authors have addressed this comment. Their explanation for why a subtraction of the fold-change values from agonist treatment alone (Fig. S9B.) to their GPCR results from Fig. 4 makes sense, though it mostly highlights a limitation of the MPRAanalyze software package rather than a limitation of their experimental methodology and technical capabilities. However, the additional supplementary volcano plot showing the fold changes across these two conditions conveys the same information, and it is good to see that the same TFs/promoter types that activated in their original analysis show up even in Fig. S9B.

The authors have adequately addressed all of my minor comments.

Reviewer #3

(Remarks to the Author)

After carefully reviewing the revised manuscript, I am delighted to confirm that the authors, Zahm et al., have sufficiently addressed all concerns raised in my initial evaluation. The revisions have significantly enhanced the clarity, coherence, and overall quality of the manuscript. Consequently, I am pleased to recommend acceptance of their study, "A Massively Parallel Reporter Assay Library to Screen Short Synthetic Promoters in Mammalian Cells" (Zahm et al., 2024), for publication.

GENERAL COMMENTS TO REVIEWERS

We would like to first thank the reviewers for volunteering their time to thoroughly review our manuscript. We are grateful for their overall positive endorsement of the work and that the value of the method was well received. We suspect many of the insightful criticisms of the manuscript (reproduced below in italics), stem from our failure to adequately articulate how our library functions and its value as a tool discovery platform. We have attempted to clarify these points both in our response and within the manuscript. Generally speaking, we believe our MPRA library presents a powerful, first-pass approach for the identification of candidate synthetic promoters. Much of that power results from the library's scale and diversity. Any candidates identified using this large-scale, high-throughput library format should then be further tested on an individual basis using orthogonal assays and incorporating additional treatment conditions such as multiple time points, drug doses, receptor antagonism, etc. Our study does include validation experiments of this fashion, although we do not extensively validate every screen hit for a specific downstream application. We instead focused our efforts on showcasing the library's potential breadth using a variety of experimental contexts, with a focus on G-protein coupled receptor signal transduction both for selective responses among receptors and signaling pathway identification. Inevitably, investigators implementing our library will desire synthetic promoters with properties specific to their needs and so will implement validation experiments geared toward assessing those properties. We have attempted to better present our reasoning and approach in various places throughout the edited manuscript, as detailed below in responses to specific comments. Ultimately, we endeavored to showcase how a user could identify and validate a desired reporter using this library. Based on reviewer feedback, we believe we've been able to substantially improve the manuscript's quality and clarity.

Note: We have modified the formatting of certain reviewer comments to better address each point, but we have not altered the text.

REVIEWER COMMENTS

Reviewer #2 (Remarks to the Author):

Summary

In Zahm et al, 2024, the authors describe = development of a massively parallel reporter assay (MPRA) for screening a library of >6,000 transcriptional reporters consisting of a combination of transcription factor binding sites, spacer lengths, DNA helix rotations, and minimal promoters using RNA abundance measurements from barcoded reporters. The authors assay the library under different culture conditions, such as FBS and forskolin, and observe responses from TFs in the library that are consistent with existing literature. They also test the library in multiple cell lines, identifying TFs that generate cell-type/cell-line specific transcriptional responses. Finally, they demonstrate the therapeutic potential of the library by assaying it against several known GPCR + agonist combinations, finding variants that respond differentially to these targets, and elucidate discrete signaling pathways associated with specific GPCRs.

Overall, this manuscript is well-written, and the results are clearly presented, with rigorous technical analysis of their results and replicate experiments that demonstrate the robustness of their method. I think it is entirely appropriate for publication in Nature Communications. However, the authors should address the following comments before the manuscript is published:

Major Comments

1. The authors should clarify the language in the text describing the purpose and utility of their study. The abstract refers to the utility of a "massively parallel reporter assay platform", that hints at a generalizability of their approach to other contexts. However, the text, and particularly the discussion, only argue for the utility the specific library used in the manuscript. When the authors describe their utility of their work, are they refereeing to the specific methodology used, i.e., cloning of a barcoded library of diverse transcriptional reporters and assaying using RNA sequencing, or to the specific set of variants (TFBSs, minimal promoters etc.) that they used in their experiments? If it is the former, the authors should describe how their platform differs from pre-existing methods (such as STARR-seq, exp-STARR-seq etc.) and what advantages it offers. If it is the latter, the language, particularly in the abstract, should be modified.

This feedback is valuable as it is clear we have failed to adequately articulate the context and value of our platform. We apologize for the confusing language used in our abstract. We indeed intended to highlight the reviewer's latter scenario - the novel utility of our approach lies in the large-scale library of specific synthetic promoters we developed, as opposed to its MPRA format. We have modified the second sentence of the abstract as follows to better emphasize this point:

“We introduce a library comprising 6,144 synthetic promoters, each shorter than 250 bp, designed as transcriptional readouts of cellular stimulus responses in massively parallel reporter assay format.”

We have also updated the third abstract sentence, substituting “library” for “format” and added “...from our promoters...” such that the sentence now reads:

“This library facilitates precise detection and amplification of transcriptional activity from our promoters, enabling the systematic development of tunable reporters with dynamic ranges of 50-100 fold.”

In the fourth, fifth, and seventh sentences of the abstract, we replaced “platform” with “library” to again emphasize the study’s utility lies in the library of novel synthetic promoters as opposed to the MPRA format itself. In the sixth abstract sentence, we replaced “technology” with “tool”. We also made similar alterations to the second and third sentence of the discussion section to reiterate this point.

Overall, our platform provides a method for identifying synthetic TRE promoters. These sequences do not exist in nature, in organism genomes, or in relevant and identifiable contexts outside our library. These sequences were purpose-built from ex vivo data and intuitions of synthetic reporter construction. As such the goal of the platform is to deliver a means for any user to survey all of our constructed reporters to identify those with utility in their own applications.

2. *Related to point 1, the authors should provide a more comprehensive discussion of the library design choices they made.*

- *For instance, Lines 294-295 state, “Importantly, this platform enables a unique method for identifying endogenously coupled synthetic reporters for 295 any signal input”.*

Again, thank you for assisting us with clarifying the intent, value, and application of our work. Our wording here was confusing. We didn’t intend to imply our library will capture all possible transcriptional outputs, rather that users could perform any treatments/studies they would like using this library (assuming compatibility with transfection and RNA isolation after treatment). The sentence in question has been modified to read:

“Nevertheless, this library provides a unique resource to identify potential endogenously-coupled synthetic reporters for all experimental designs compatible with plasmid delivery and barcode recovery.”

- *While the authors describe developing their library from HT-SELEX, it appears they sampled only a subset of the variants from the cited paper. It is not clearly communicated how this subset was chosen. Moreover, it is unclear if these TREs are chosen for diversity of cell signal detection.*

The information regarding the TF (motif) selection process we used is described within the methods section title “Transcription factor binding motif selection”. We have attempted to better explain the process with the changes below. Importantly, we did not remove any motifs derived from the position weight matrices of the referenced HT-SELEX study. The number of motifs we included in the library (325) is lower than the number of transcription factors included in the HT-SELEX experiments (891 human and 444 mouse TF DNA-binding domains and 984 full length human TFs) only because they only obtained specific sequences for a subset of these (411 total), from which we then removed redundancy (i.e., we removed any motif that was the same as, or entirely present within, another motif. For example, if our position weight matrix processing described in the method section produced the motifs “CAAAAAC” and “AAAAA”, we only designed promoters based on “CAAAAAC”). We have attempted to clarify this point by making two additions. We have added “all unique” to the second sentence of the first results section, such that it reads:

“We developed a barcoded plasmid library (‘TRE-MPRA’) of synthetic promoters composed of TF binding motifs derived from **all unique** DNA position weight matrices for hundreds of human and mouse TFs identified via HT-SELEX (**see Materials and Methods**).”

We also added wording to the second sentence of the relevant methods section so that it reads:

“Beginning with the seed sequence of each PWM in the HT-SELEX dataset, we first trimmed fully degenerate nucleotides (Ns) from the 5’ and 3’ ends...”

- *If their approach is indeed generalizable to other library compositions and experimental contexts, it would be helpful to understand the motivations for how they chose their library and what classes of problems they expect their platform to address.*

Our library composition was limited as far as TF/TRE representation due only to the HT-SELEX dataset we opted to use. Our rationale for using this particular type of data is presented in the first paragraph of the results section (third sentence):

“We reasoned that TREs based on DNA sequences bound by TFs *ex vivo* would produce superior synthetic promoters compared to sequences based on genomic footprints and removed from their native chromatin context.”

The experiment from Jolme et al. was, and likely still is, the largest HT-SELEX experiment as far as the number of transcription factors assayed. As noted above, we included all possible TREs generated by that dataset. However, the TRE-MPRA library we introduce here could be supplemented or remade to include candidate TREs from additional transcription factors, whether the TREs were determined via SELEX or other methodologies, and so our approach should be viewed as generalizable in that respect. We have added the following to the last paragraph of the discussion to relay this idea:

“Furthermore, our MPRA plasmid library can be easily supplemented with new synthetic promoters based on the binding motifs of additional TFs, as warranted.”

Regarding the classes of problems our platform could be used to address, we have rewritten a portion of the discussion in the hopes of clarifying our vision regarding its wide application:

“Nevertheless, this library provides a unique resource to identify potential endogenously-coupled synthetic reporters for all experimental designs compatible with plasmid delivery and barcode recovery. We anticipate this system being of significant utility in applications where synthetic circuit design, drug target identity, or biological effectors of cellular response are required, but unknown, due to novel biological context or condition. Users can directly screen this library, identify reporters ready for immediate use, and leverage them for a myriad of applications including deep mutational scanning, preclinical therapeutic monitoring, and the control of chimeric antigen receptor T cells *in vivo*.”

- *It would be useful to see an overview, perhaps in figure form, that uses insights from their experiments to help the reader understand, e.g., how many TREs, min promoters, number of barcodes, sequencing depth, etc. should be considered for deployment of this platform in for other applications in other experimental contexts.*

Thank you for this suggestion. It again appears we have not sufficiently articulated the context, application, and value proposition of this platform adequately. The library we present here cannot be tailored in its present form to reduce its scale, as the barcoded plasmids are already pooled. A reduction in TREs, minimal promoters, and/or barcodes would require the investigator(s) to re-synthesize a smaller scale pool of plasmids. However, the library in its present form can be administered to any transfectable cell backgrounds and stimulated in any context, format, or with any stimulus or set of additional conditions. This was not explicitly stated and we now mention in the discussion section that our library can be applied in any desired experimental context.

As far as guidance toward deployment of the current library, we have now included a recommendation in the methods section “Processing of sequencing data” relating to scales we think should be targeted using this library:

“Note: based on the barcode recovery and sequencing depth obtained in this study, we recommend using similar cell numbers, transfection methods, and sequencing library preparation methods, and to sequence at similar depth to help ensure data quality.”

3. GPCR results observed in Fig. 4 should be contrasted/normalized with those observed in Fig. SB. In Fig 4, the library is transiently transfected along with a GPCR and cognate small molecule agonists, and the fold change in transcriptional

output is computed as the ratio of the GPCR + agonist counts over the GPCR counts alone. However, the results in Fig S9B show that there is an effect observed by simply treating the cells with the agonist without transfecting the cognate GPCR, which the authors attribute to endogenous GPCR signaling. Given this result and hypothesis, this effect should be subtracted from the values obtained in Fig. 4A to distinguish endogenous GPCR activation from the effects of the transiently transfected GPCRs to clearly distinguish the two.

Unfortunately, we are unable to subtract promoter values of epinephrine treated cells from those values presented in Figure 4A, as our pipeline utilizes raw sequencing read counts as input and generates models for likelihood ratio tests using the MPRAanalyze software package. MPRAanalyze can generate models for experiments with more than two groups, yet we chose not to present these types of analyses because the output simply identifies significantly-altered promoters without supplying fold changes (direction of magnitude) or pairwise significance tests between individual sample types (similar to an ANOVA without post-hoc testing). We felt a simple list of p-values generated by a comparison with more than two groups was less informative than the pairwise comparisons we performed and presented in volcano plots.

However, we certainly agree with the reviewer that this remains an important comparison to make, one we had done but not fully shared in the original manuscript. We have added a new volcano plot depicting the fold changes and significance for promoters in cells treated with epinephrine alone versus cells treated with epinephrine and the ADRB2 expression plasmid to Figure S9B (and reproduced below for reference). These fold changes represent differential promoter activity between overexpressed ADRB2 in the presence of epinephrine versus endogenous GPCRs in the presence of epinephrine. Figure 4C displays the activation of two promoter types (CRE and Mafb) across the four treatment groups of the ADRB2 experiment, which serves to consolidate the findings of the ADRB2 pairwise comparisons, including that of the newly-added volcano plot, and perhaps helps the reader distinguish between effects of agonist treatment alone and signaling from transiently overexpressed ADRB2.

The reviewer's point regarding the effect of agonist alone on promoter activities is something we considered carefully when designing our experiments. We performed these experiments explicitly to assess whether endogenous receptor activation by ligand, or overexpression of an endogenously present ligand, presented a greater impact on background signal accumulation. As observed in the figure comparing ADRB2 endogenous and overexpressed states +/- agonist, overexpression of the receptor itself produced a higher baseline signal than fully agonized endogenous receptor. As such we chose overexpressed receptor signal as our baseline comparison for agonist mediated signal activity in our pairwise comparisons. Although we did include an agonist-only condition in the ADRB2 experiment as a proof-of-concept, we felt our resources (i.e. sequencing reads) were best spent on assaying more GPCRs and other treatments rather than agonist-only conditions, which could then be included in validation experiments. To demonstrate this type of orthogonal validation process, we presented a luciferase experiment incorporating agonist-only controls in the GPR91-responsive promoter validation process in Figure S10B. Our ultimate goal in the TRE MPRA experiments is to identify synthetic reporters responsive to the agonist treated condition to acquire synthetic readouts of activity for the target receptor, or unique to that particular receptor among GPCRs of classically similar coupling identity. Follow-up validation will always be essential in these contexts.

Minor Comments

1. Fig. S2B does not mention which correlation coefficient is shown in the plots (I'm assuming it's spearman's ρ), and should be added to the figure legend

The reviewer is correct, the coefficients presented in Figure S2B are Spearman's rho. We also made the same omission for Figure S2D. We have updated the legends for each to specify the coefficient used.

2. Page 2, lines 82-83, "...whereas the spacer sequences between TF binding motifs and the distance between the TRE unit and minimal promoter did not globally alter transcription". The authors should mention if the parameters relevant to rotation/spacing they considered in this study were not sufficient to cause a significant change in expression, or that these features do not dictate transcription rate at large. The authors should also offer some discussion on the utility and importance of testing different rotations, spacers, etc

While we did not observe a global effect on transcription caused by the nucleotide distance (spacing) between the TRE and minimal promoter, we did present spacing effects for some promoters in FBS and forskolin experiments (Figure S4C). Notably, the specific ETS1-based promoter shown in Figure S4C also shows similar spacer effects following agonism of certain GPCRs (when ETS1 responded to GPCR agonism, this spacer effect was also present). Likewise, the GMEB2-based promoter shown in Figure S4C also showed similar spacer effects following treatment with ATP/GTP as it did with forskolin. These observations suggest that spacing of these individual promoters is important for the transcriptional machinery involved in their activation, independent of the stimulus. We have added the following sentence to the end of the paragraph referenced here by the reviewer:

"This result suggests that, while not an important consideration for many TREs, the distance between TRE and paired minimal promoter significantly affects synthetic promoter utilization by certain TFs."

3. In Fig. 2D, F, why were the specific choices of rotations to plot used in these plots (ie rotation 8 for THRB-1, set 1, for 2D, and rotation 1 for Mafb, set 1 for 2F)? Do the other rotations not show these effects, or was this one chosen at random?

The choice here was random. All promoters with THRB-1 (FBS) or Mafb (Forskolin) had FDR values of 0 when compared to untreated controls in our MPRA experiments. Neither of the selected architectures had the highest LRT statistics among promoters containing the same TREs (i.e. were not the highest on the y-axes of Figure 2A for their TRE). MPRA fold changes were comparable across spacers and rotations with the exception of THRB-1 rotation 4, which had slightly lower (but still highly significant!) fold changes when paired with the Promega or thymidine kinase minimal promoter.

To be clear, these were the only versions of THRB-1 and Mafb promoters we tested via luciferase assay in this experiment (we didn't omit other results). Because each promoter we tested orthogonally in this study well recapitulated the results of our MPRA experiments, we are confident that the other THRB-1 and Mafb promoter architectures (MPRA results for all of these are displayed in Figures 2A, 2C and 2E) would have also responded to FBS and Forskolin in a dose-dependent manner, had they been tested in this format. As the reviewer points out, it would be prudent to specify a reason for selecting these particular promoters for validation in the text. We have added the following wording:

"The promoters selected for validation reflected the general MPRA responses of all promoters containing these TREs under these treatment conditions."

4. Page 3, Line 117-118, "This discordant finding may reflect TF/TRE-specific translation rates observed in previous studies, and emphasizes the necessity to validate individual candidate promoters in orthogonal assays". This is a very salient and important point made by the authors. MPRA results should often be validated in a secondary experiment, especially when screening in a different context!

5. In Fig. 3D, S7B, when testing the library against heavy metals, were any cell health effects or toxicity observed? The authors should consider showing that there were no effects on cell health, especially due to there being only a single replicate used in these assays. If any effects of cell health are observed, the authors should comment on the same

After six hours of treatment with zinc, HEK293 cells had acquired a rounded shape relative to untreated and cadmium treated cells, consistent with toxicity. Notably, the cells remained attached to the tissue culture flask and were washed prior to trypsinization without noticeable detachment. The RNA yield, library preparation and

sequencing depth/quality from this sample were on par with the other treatments so we don't believe that the clearly developing toxic effect hampered promoter activity from 0-6 hours, at least in a way that compromised our ability to collect barcoded mRNA accumulated during that time. We did not observe any gross differences in cell health for any of the other conditions in our MPRA experiments, including in cells treated with thapsigargin, although we didn't perform cell viability or vitality assays. Not surprisingly, there appeared to be more cells on the plates treated with FBS for six hours, although we didn't quantify this observation. We have added light microscope images of vehicle-, zinc-, and cadmium-treated HEK293 cells at the six hour time point as Figure S7C (and below for reference) and have amended the relevant results section to including the following wording:

"We noted that both metal treatments induced similar responses in metal response element (MRE)-containing and Tp53-containing promoters, relative to vehicle-treated controls, **despite the apparent toxicity of zinc but not cadmium** (Fig. 3D, S7B,C)."

6. Page 4, Line 179, potential typo - "single", instead of "signal"

Yes, this was indeed a typo. We intended to use 'single'. We have updated the text accordingly.

7. Page 5, Line 267, "...promoters with the highest dynamic-range yet generated for these applications". This claim should be coupled with potential caveats of using a luciferase based reporter vs other, more commonly used outputs such as fluorescent proteins e.g. EGFP. Assays and resulting fold changes from an enzymatic reporter cannot be directly compared against FP based outputs due to the significant differences in their mechanism

This is absolutely correct and the breadth of this statement makes the intention unclear. Enzymatic assays cannot be correlated on magnitude to fluorescent or other linear assay readouts. In addition, we did not validate all of our reporters for translational response. The intention here was purely from the standpoint of the transcriptional readouts of these synthetic reporters within the MPRA platform. We have edited the sentence as follows:

"endogenously-coupled functional promoters with the highest **transcriptional** dynamic range yet generated for these applications."

Reviewer #3 (Remarks to the Author):

The author presents a highly efficient, massively parallel reporter assay platform of synthetic promoters, designed to assess cellular responses to stimuli quantitatively. As the author suggests, this platform enables precise transcriptional activity detection and amplification, facilitating the development of tuneable reporters with dynamic ranges of 50-100 fold. Demonstrating broad applicability, the platform functions across multiple cell lines and responds to various stimuli, including metabolites, mitogens, toxins, and pharmaceutical agents. Notably, the authors also demonstrate its utility in therapeutic development by generating unique reporters for drug target signals, particularly for G-protein coupled receptors (GPCRs), critical in drug discovery. The platform's ability to isolate and define specific signaling pathways associated with which could be valuable tool in synthetic biology, cellular engineering, ligand exploration, and drug development.

Overall the paper is well written with well-designed and sufficient experiments but quite a few concerns need to be addressed.

The paper mentions existing technologies like STARR-seq and other MPRA's but does not adequately compare them to the new approach introduced in this study. A more detailed comparison would help in highlighting the novelty and potential advantages of the presented library.

We have now included some additional text to the introduction in an effort to contrast the referenced STARR-seq and MPRA's with our plasmid library. The point we are attempting to convey is that we have borrowed the referenced experimental designs (i.e. using barcodes as readouts of gene regulation at very high throughput) previously used to interrogate genomic promoter and enhancer sequences and applied this format to a tool for screening synthetic promoters we have generated based on sequences bound by TFs *ex vivo*. The relevant text of the final paragraph of the introduction now reads:

“Technologies such as self-transcribing active regulatory region sequencing (STARR-seq) and massively parallel reporter assays (MPRA's) have been used to interrogate the effects of primary DNA sequence on gene expression at extremely high throughput. These studies have finely mapped many rules governing the highly complex TF-DNA interactions at the heart of gene regulation. Here, we leverage the experimental principles of these high throughput formats to create a powerful tool to easily quantify responses of novel synthetic promoters that serve as downstream transcriptional readouts of specific upstream signaling events in mammalian cells. Our MPRA plasmid library can be used to survey over five hundred thousand barcoded plasmids representing 6144 synthetic promoters of less than 250 bp in length, containing candidate TREs derived from the *ex vivo* binding motifs of 229 human and mouse TFs.”

The discussion does not address the potential limitations of the approach for the synthetic promoters being studied. Acknowledging these upfront would provide a more balanced view of the research and its potential challenges.

This is a very good suggestion; our manuscript should absolutely discuss limitations of our library and its format. We have added the following new paragraph to the discussion section to address some of these limitations:

“Although we have demonstrated the value of our library in multiple contexts, several limitations inherent in our library composition warrant consideration. First, the library almost certainly is not able to capture the activity of every mammalian transcription factor. The HT-SELEX study upon which our synthetic promoters was based did not obtain position weight matrices for every human and mouse transcription factor and so our library will not provide direct readouts of their activities. It is plausible that synthetic promoters simply cannot be derived to detect certain TFs, which may lack functionality except upon native chromatin. In addition, our library may not include the spatial architectures necessary for certain TFs to regulate transcription from the minimal promoters. Promoter architecture diversity is a delicate design consideration, as including too few varieties may inadvertently render the library insensitive to certain TFs, whereas including more increases the library diversity and subsequently the experimental scale necessary to obtain library coverage without a similar gain in information. Our library has likely skewed toward the latter, as many TREs displayed a similar response across each configuration. Furthermore, as exemplified by our G α_i -coupled GPCR activation experiments, certain biological responses will not be reflected by changes in the output of our synthetic promoters. One inherent shortfall in the MPRA format is lack of temporal information, as each sample will include barcodes expressed beginning from the time of transfection up to the time of collection. Because of the costs associated with NGS, we recommend temporal resolution be expanded during promoter validation.”

The paper does not fully address potential non-specific effects of treatments, which could confound the results and compromise the reliability of the conclusions. Specifically, the study lacks controls for non-specific effects in experiments using fetal bovine serum (FBS), forskolin, zinc, and cadmium, potentially leading to misinterpretation of promoter activity changes.

We thank the reviewer for this perspective and strongly agree with the value of multiple controls and the necessity of these conditions across experimental approaches. Perhaps most valuable in the context of our work is to further highlight the throughput constraints of conditions in our platform. Given the scale of the library and the depth of reads necessary per condition a maximum of 40 conditions can be run on the NovaSeq 6000 SP flow cells used in this study. As a result, we ran control conditions as suggested for select treatments to evaluate cost/benefit of dedicating read depth to specific experimental conditions vs. increasing the breadth of our analysis across cell types and treatment conditions.

- *For example: Line 84-100: The paper describes treating HEK293 cells with fetal bovine serum (FBS) and forskolin to benchmark the synthetic promoters but does not mention controls for non-specific effects. This lack of controls could lead to misinterpretation of the promoter activity changes.*

We have corrected this omission within the paragraph of lines 84-100 by replacing the term “untreated” with “vehicle-treated” to clarify the control treatment conditions.

- *Line 137-144: When discussing heavy metal treatments (zinc and cadmium), the authors note differential promoter responses but do not explicitly discuss controls for potential non-specific effects of these treatments.*

Cells treated with the heavy metals, and all other treatments with compounds (with the exception of FBS), were compared to vehicle-treated cells. FBS-treated cells were compared to cells in serum-free media. We have now modified the referenced results section (lines 142-3) to correct our omission:

“We noted that both metal treatments induced similar responses in metal response element (MRE)-containing and Tp53-containing promoters, **relative to vehicle-treated controls** (Fig. 3D, S7B).”

- *Figures illustrating promoter responses to these treatments also omit necessary controls, such as vehicle-only conditions, which are crucial for distinguishing specific from non-specific effects.*

We have updated figure axes to more accurately reflect the specific comparisons being displayed. The term “Untreated” has been replaced with the more precise “Vehicle” and “Serum Free” designations.

- *Also including antagonists for aminergic receptors as a parallel condition could have helped delineated non-specific further more. Including such controls and measuring off-target effects would improve the accuracy and interpretation of the synthetic promoter responses.*

When designing our MPRA treatment conditions, we carefully weighed the utility of GPCR antagonists against the sequencing read depth necessary for each additional treatment condition. Often, antagonists of GPCRs actually function as partial agonists or inverse agonists, and many are known to have off-target/non-specific effects, which may have confounded results from the MPRA format further. We concluded that experiments including antagonists, while likely informative at the level of MPRA, were best left to the candidate promoter validation phases from a cost-benefit perspective.

In fact, for at least one of the GPCR a dose-dependent readout of the promoter activity readout also may better reflect usability of these tool sets.

We performed a handful of orthogonal dose-dependent luciferase assays using GPCRs tested with our MPRA platform, although not all. Figure 5E shows dose-response curves for select promoters with agonism of NTSR1 and HTR2A. Figure S10B shows dose-response curves for a promoter activated by GPR91 agonized with cis-epoxysuccinate. We also included a dose-curve for cis-epoxysuccinate treatment in cells transfected with a control vector to demonstrate the promoter activation is dependent upon GPR91 expression. All of these conditions also include a no drug control.

The study's measurements were taken at a single time point, which may not fully capture the dynamics of GPCR signaling and promoter activation. This approach risks missing important transient or delayed responses, leading to an incomplete understanding of promoter activities. For instance, in line 75-80, The paper mentions transfecting HEK293 cells and measuring promoter activities after a specific treatment but does not detail multiple time points. And in line 156-163, author mentions serum responses were measured at a single time point (6 hours) across different cell lines, revealing discordant responses but lacking temporal resolution. Comparing promoter rankings and responses to fetal bovine serum (FBS) also reflects this limitation. Including multiple time points in the experimental design would provide a more comprehensive view of promoter activation, revealing early, intermediate, and late transcriptional responses that will certainly improve the temporal resolution of GPCR signaling dynamics.

The reviewer is certainly correct, the single time point we used for our MPRA experiments does indeed fail to capture promoter activity dynamics occurring from the period of treatment initiation (hour 0) to RNA harvest (hour 6). We believe our data does capture any promoters activated very early and then possibly turned back off prior to

hour 6, so we are confident we haven't missed promoter activations of different dynamics during this time frame. We chose to stop the treatment at six hours because we wanted to avoid collecting barcodes produced by TFs becoming activated indirectly as a result of signaling cascades triggered by the initial transcriptional activity (i.e. avoid measuring delayed responses). Biologically, these delayed responses may be just as important as the initial activities. However, we feel the first wave of responding synthetic promoters are of higher value for the investigator attempting to identify a candidate synthetic promoter(s) that can be leveraged as a readout of a specific experimental stimulus. Our assay could be performed using other or additional time points, as desired, without issue. We chose to focus on a single time point in this study so that our sequencing reads could be distributed across more treatments as opposed to more time points and compared directly. The reviewer is correct that it remains important to still highlight the relevant impact of time on TRE MPRA activity and output. We have now included the duration of treatment for the baseline (serum-free) sample collection (lines 77-8) and the fetal bovine and forskolin treatments in the main text (lines 86-7).

In Lines 198-199, the authors suggest that activation of Gai-coupled GPCRs does not influence cellular transcription in HEK293 cells. However, the lack of perturbation observed with the D2R receptor in terms of promoter activity may be attributed to the dopamine concentration used in the experiment, which was 1 μ M. It is plausible that a higher concentration could induce measurable activity, as the EC50 of the endogenous ligand for the D2R receptor is within a higher ligand concentration range. Thus, the potential for transcriptional regulation at increased dopamine levels cannot be definitively ruled out. Authors may try their toolset in other Gi coupled receptors and verify this claim.

The reviewer is correct that a dose of dopamine higher than 1 μ M may have activated the D2R receptor sufficiently to generate a transcriptional response. Nevertheless, we feel our supposition that Gi-coupled GPCR activation does not affect our synthetic promoter activities in HEK293 cells is correct for a few reasons:

1. As the reviewer correctly points out, our hypothesis could be tested by running our TRE-MPRA platform using a different Gi-coupled receptor. We had precisely the same thought and that is why we performed the experiment using the mu opioid receptor, a non-aminergic GPCR that is known to predominantly couple to, and signal through, Gi. The results of this experiment are presented in the heatmap of Figure 4B and with a volcano plot in Figure S9A. Indeed, treatment with the mu opioid receptor agonist morphine in cells overexpressing this Gi-coupled receptor also showed no significant alteration of our synthetic promoter activities. From experiments using BRET-based sensors, we know that our 100 nM dose of morphine is sufficient to uncouple and active Gi from plasmid-expressed mu opioid receptors.
2. Like the reviewer, we also wondered whether 1 μ M of dopamine was a sufficient dose to activate receptor signaling leading to a transcriptional response. Thus, we attempted the same experiment with the same dose of dopamine, but replaced overexpressed D2R with overexpressed D1R. D1R is a GPCR that couples primarily to Gs as opposed to Gi. The results of this experiment are shown in the heatmap of Figure 4B and with a volcano plot in Figure S9A. Our results show that a dose of 1 μ M dopamine is sufficient to trigger transcriptional responses from D1R-Gs.
3. Previous reports utilizing BRET-based assays in HEK293 cells have shown dopamine EC50 values well below 1 μ M for D2R uncoupling of Gi (PMID: 38643909; PMID: 35403009) and D1R uncoupling of Gs (<https://doi.org/10.1101/2023.10.03.560682>). According to PDSP's Ki database, the average inhibitory constant for dopamine at D2R reported in the literature is below 1 μ M (<https://pdsp.unc.edu/databases/pdsp.php>). Thus, we feel our treatment of 1 μ M dopamine was sufficient to agonize D2R and trigger Gi uncoupling in our MPRA experiment. One important caveat regarding dopamine concentrations across the literature is the rapid oxidation of dopamine in aqueous solutions. We administer dopamine in the presence of ascorbic acid, which prevents this oxidation. Other investigators may not take care to prevent oxidation and thus require higher dopamine concentrations to achieve detectable receptor agonism.

In the experiments involving endogenous β 2AR, the observed increase in transcription from the promoters in the absence of epinephrine treatment was attributed to basal constitutive signaling, a well-documented phenomenon. While this is a plausible explanation, attributing the observed transcriptional activity specifically to autocrine signaling would be an overextension. To substantiate the claim of autocrine activity, the authors should provide supporting experimental evidence or reference similar findings from other studies. In the absence of such corroboration, it would be prudent for the authors to refrain from making this claim (line 207-208).

We agree that our suggestion of autocrine activation is an overextension based on the data collected. We have modified the relevant text to remove this suggestion. The relevant sentence now reads:

“Furthermore, the overexpression of ADRB2 in the absence of epinephrine treatment also caused increased transcription from these promoters, suggesting that our platform can detect basal constitutive signaling from plasmid-expressed GPCRs (Fig. S9B).”

Reviewer #3 (Remarks on code availability):

Details are available and are sufficiently described.

Response to Reviewers' Comments

Reviewer #2 (Remarks to the Author):

The authors have made significant changes in response to the comments I presented, improving the clarity of the manuscript, and clearly communicating the novelty of their method, which lies in the library they have developed as a means for identifying candidate transcriptional response element arrays for a range of downstream applications. They do a good job of clarifying that the library screen is meant to serve as a starting point for users, and that appropriate validation in the user-defined context of their application will be necessary to screen any "hits" returned from the initial screening of their library will be crucial, as it is an important point for most high-throughput assays used in this fashion.

Based on the authors' responses to the comments, their edits to the main text and supplement, I believe the manuscript is now fit for publication in nature communications. Detailed responses to their rebuttal of my specific comments are appended below:

Major Comments

- 1. The authors' edits to the abstract sufficiently address this comment and bring clarity to the value of their work, i.e. a large-scale exhaustive library of transcription factors, minimal promoters, and some other relevant features that is meant to serve as a first-pass screen for a variety of applications.*
- 2. The additional context added by the authors to this claim sufficiently addresses this comment. The motif selection strategy that the authors used makes sense, and the changes they have made to the material, as well as to the methods section. However, the example they have presented here where instances of one motif being a subset of another and therefore being excluded clarifies their filtering approach very intuitively, and I'd urge the authors to include this example in their methods section as well, to provide further clarity to the readers. The significant edits made to the text describing potential use cases for their library also brings clarity. Further, the overarching clarification of the goal of this work being the development of this library, which wasn't immediately apparent during the initial submission coupled with these edits have simplified the nature of the discussion necessary with regard to this comment. If this work served as more general platform for RNA-sequencing based readouts of pooled variants using barcode libraries, there would be additional clarification necessary, but the authors have clarified that that is not the case with this work. The recommendations the author make in the methods section is an important addition to the manuscript. However, I'd encourage the authors to include specific numbers regarding their sequencing read depth and any other relevant metrics in their methods section, to gain further insight into the specifics of their results.*

We have now updated the relevant portion of the Methods section titled 'Transcription factor binding motif selection' to read as follows based on the reviewer's suggestion:

"Finally, to eliminate redundancy in our candidate list, we removed any resulting motif for which the entire sequence was represented within another motif, resulting in a total of 325 unique binding motifs (Supplementary Table 2). For example, if our position weight matrix processing produced the motifs CAAAAAC and AAAAA, we only designed promoters based on the CAAAAAC motif."

- 3. The authors have addressed this comment. Their explanation for why a subtraction of the fold-change values from agonist treatment alone (Fig. S9B.) to their GPCR results from Fig. 4 makes sense, though it mostly highlights a limitation of the MPRAanalyze software package rather than a limitation of their experimental methodology and technical capabilities. However, the additional supplementary volcano plot showing the fold changes across these two conditions conveys the same information, and it is good to see that the same TFs/promoter types that activated in their original analysis show up even in Fig. S9B.*

The authors have adequately addressed all of my minor comments.

Reviewer #3 (Remarks to the Author):

After carefully reviewing the revised manuscript, I am delighted to confirm that the authors, Zahm et al., have sufficiently addressed all concerns raised in my initial evaluation. The revisions have significantly enhanced the clarity, coherence, and overall quality of the manuscript. Consequently, I am pleased to recommend acceptance of their study, "A Massively Parallel Reporter Assay Library to Screen Short Synthetic Promoters in Mammalian Cells" (Zahm et al., 2024), for publication.

Reviewer #3 (Remarks on code availability):

Unfortunately, I am not an expert in codes. Please take suggestion for some expert for this part.